# Large Language Diffusion Models

**Shen Nie**[1,2,3*†] **Fengqi Zhu**[1,2,3*†] **Zebin You**[1,2,3†] **Xiaolu Zhang**[4‡] **Jingyang Ou**[1,2,3]
**Jun Hu**[4‡] **Jun Zhou**[4] **Yankai Lin**[1,2,3‡] **Ji-Rong Wen**[1,2,3] **Chongxuan Li**[1,2,3‡§]

[1] Gaoling School of Artificial Intelligence, Renmin University of China
[2] Beijing Key Laboratory of Research on Large Models and Intelligent Governance
[3] Engineering Research Center of Next-Generation Intelligent Search and Recommendation, MOE
[4] Ant Group
{nieshen,fengqizhu,chongxuanli}@ruc.edu.cn

## Abstract

The capabilities of large language models (LLMs) are widely regarded as relying on autoregressive models (ARMs). We challenge this notion by introducing *LLaDA*, a diffusion model trained from scratch under the pre-training and supervised fine-tuning (SFT) paradigm. LLaDA employs a forward data masking process and a reverse generation process, parameterized by a Transformer to predict masked tokens. It provides a principled generative approach for probabilistic inference by optimizing a likelihood lower bound. Across extensive benchmarks on general tasks, math, code, and so on, LLaDA demonstrates strong *scalability* and performs comparably to our self-constructed ARM baselines. Remarkably, LLaDA 8B is competitive with strong LLMs like LLaMA3 8B in *in-context learning* and, after SFT, exhibits impressive *instruction-following* abilities in case studies such as multi-turn dialogue. Moreover, LLaDA addresses the reversal curse, surpassing GPT-4o in a reversal poem completion task. Our findings show the promise of diffusion models for language modeling at scale and challenge the common assumption that core LLM capabilities discussed above inherently depend on ARMs. Project page and codes: `https://ml-gsai.github.io/LLaDA-demo/`.

## 1 Introduction

Large language models (LLMs) [1] fall entirely within the framework of generative modeling. Specifically, LLMs aim to capture the true but unknown language distribution $p_{\text{data}}(\cdot)$ by optimizing a model distribution $p_\theta(\cdot)$ through maximum likelihood estimation, or equivalently KL divergence minimization between the two distributions:

$$\underbrace{\max_\theta \mathbb{E}_{p_{\text{data}}(x)} \log p_\theta(x) \Leftrightarrow \min_\theta \text{KL}(p_{\text{data}}(x)||p_\theta(x))}_{\text{Generative modeling principles}}. \tag{1}$$

The predominant approach relies on the autoregressive modeling (ARM)—commonly referred to as the "next-token prediction" paradigm—to define the model distribution:

$$\underbrace{p_\theta(x) = p_\theta(x^1) \prod_{i=2}^{L} p_\theta(x^i \mid x^1, \ldots, x^{i-1})}_{\text{Autoregressive formulation}}, \tag{2}$$

---

[*]Equal contribution.

[†]Work done during an internship at Ant Group.

[‡]Project leaders.

[§]Correspondence to Chongxuan Li.

39th Conference on Neural Information Processing Systems (NeurIPS 2025).

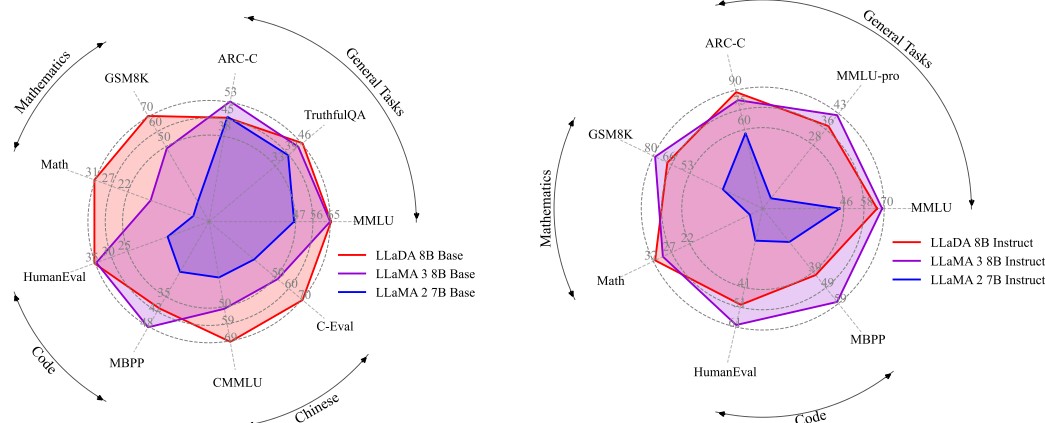

Figure 1: **Zero/Few-Shot Benchmarks.** We scale LLaDA to 8B parameters from scratch and observe competitive zero/few-shot performance compared with strong autoregressive LLMs [6].

where $x$ is a sequence of length $L$, and $x^i$ is the $i$-th token. This paradigm has proven remarkably effective [2–5] and has become the foundation of current LLMs. Despite its widespread adoption, a fundamental question remains unanswered: *Is the autoregressive paradigm the only path to achieving the core capabilities of LLMs, such as scalability, in-context learning, and instruction-following?*

We argue that the answer is *not* a simple "yes". The key insight overlooked previously is: It is the *generative modeling principles* (i.e., Eq. (1)), *rather than the autoregressive formulation* (i.e., Eq. (2)) itself, that fundamentally underpin the essential properties of LLMs.

In particular, we argue that *scalability* is primarily a consequence of the interplay between Transformers [7], model size, data size, and *Fisher consistency*[5] [8] induced by the generative principles in Eq. (1), rather than a unique result of the ARMs in Eq. (2). The success of diffusion transformers [9, 10] on visual data [11] supports this claim. Furthermore, the *instruction-following* and *in-context learning* [4] capabilities appear to be intrinsic properties of all conditional generative models on structurally consistent linguistic tasks, rather than exclusive advantages of ARMs. In addition, while ARMs can be interpreted as a *lossless data compressor* [12, 13], any sufficiently expressive probabilistic model can achieve similar capabilities [14].

However, certain inherent limitations of LLMs can be directly attributed to their autoregressive nature. For instance, the left-to-right generation process restricts their ability to handle reversal reasoning tasks [15], highlighting a representative failure in the generalization capabilities of current models.

Motivated by these insights, we introduce *LLaDA (Large Language Diffusion with mAsking)* to investigate whether the capabilities exhibited by LLMs can emerge from generative modeling principles beyond ARMs, thereby addressing the fundamental question posed earlier. In contrast to traditional ARMs, LLaDA leverages a masked diffusion model (MDM) [16–20], which incorporates a forward data masking process and trains a *mask predictor* to approximate its reverse process. This design enables LLaDA to construct a model distribution with bidirectional dependencies and optimize a variational lower bound of its log-likelihood, offering a principled and previously unexplored perspective on the core capabilities of LLMs discussed above.

We adopt the standard pipeline of data preparation, pre-training, supervised fine-tuning (SFT), and evaluation, scaling LLaDA to an unprecedented language diffusion of size 8B. In particular, LLaDA 8B was pre-trained from scratch on 2.3 trillion tokens using 0.13 million H800 GPU hours, followed by SFT on 4.5 million pairs. Across diverse tasks, including language understanding, math, code, and Chinese, LLaDA demonstrates the following contributions:

- LLaDA scales effectively to a compute budget of $10^{23}$ FLOPs, achieving comparable results to ARM baselines trained on the same data across six tasks, e.g., MMLU and GSM8K.

---

[5]It suggests the ability to recover the true data distribution with infinite data, a sufficiently large network and optimal training.

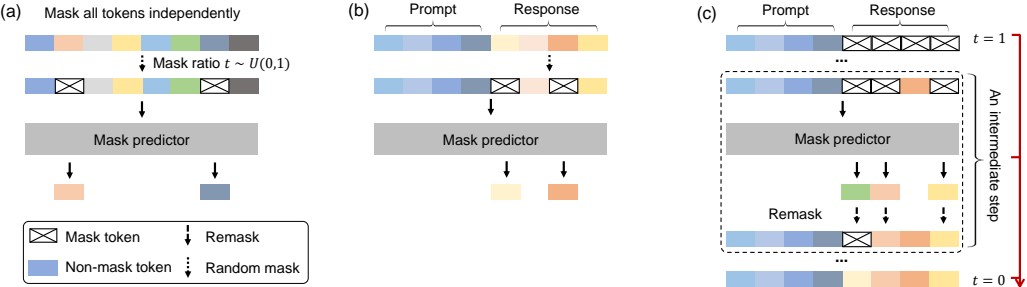

Figure 2: **Overview of LLaDA.** (a) Pre-training. LLaDA is trained on text with random masks applied independently to all tokens at the same ratio $t \sim U[0, 1]$. (b) SFT. Only response tokens are possibly masked. (c) Sampling. LLaDA simulates a diffusion process from $t = 1$ (fully masked) to $t = 0$ (unmasked), predicting all masks simultaneously at each step with flexible remask strategies.

- The pre-trained LLaDA 8B Base surpasses LLaMA2 7B Base [21] on nearly all 15 standard zero/few-shot learning tasks while performing on par with LLaMA3 8B Base [6], showcasing effective in-context learning capability.
- LLaDA significantly enhances the ability to follow instructions after SFT, as demonstrated in case studies such as multi-turn dialogue.
- LLaDA effectively breaks the reversal curse [15] with consistent performance across forward and reversal tasks. Notably, it outperforms GPT-4o in a reversal poem completion task.

## 2 Approach

In this section, we introduce the probabilistic formulation[6], along with the pre-training, supervised fine-tuning, and inference procedures for LLaDA, as illustrated in Fig. 2.

### 2.1 Probabilistic Formulation

Unlike ARMs in Eq. (2), LLaDA defines a model distribution $p_\theta(x_0)$ through a *forward process* and a *reverse process* [16–20]. The forward process gradually masks tokens independently in $x_0$ until the sequence is fully masked at $t = 1$. For $t \in (0, 1)$, the sequence $x_t$ is partially masked, with each being masked with probability $t$ or remaining unmasked with probability $1 - t$. The reverse process recovers the data distribution by iteratively predicting masked tokens as $t$ moves from 1 to 0.

The core of LLaDA is a *mask predictor*, a parametric model $p_\theta(\cdot|x_t)$ that takes $x_t$ as input and predicts all masked tokens (denoted as M) simultaneously. It is trained using a cross-entropy loss computed only on the masked tokens [18–20]:

$$\mathcal{L}(\theta) \triangleq -\mathbb{E}_{t,x_0,x_t} \left[ \frac{1}{t} \sum_{i=1}^{L} \mathbf{1}[x_t^i = \mathrm{M}] \log p_\theta(x_0^i|x_t) \right], \tag{3}$$

where $x_0$ is a training sample, $t$ is a continuous random variable drawn uniformly from $[0, 1]$, $x_t$ is sampled from the forward process and $L$ is the sequence length. The indicator function $\mathbf{1}[\cdot]$ ensures that the loss is computed only for masked tokens.

Once trained, we can simulate a reverse process (see Sec. 2.4 for details) parameterized by the mask predictor and define the model distribution $p_\theta(x_0)$ as the marginal distribution induced at $t = 0$. The loss function in Eq. (3) has been proven to be an upper bound on the negative log-likelihood of the model distribution, making it a principled objective for generative modeling:

$$-\mathbb{E}_{p_{\text{data}}(x_0)} \left[ \log p_\theta(x_0) \right] \leq \mathcal{L}(\theta). \tag{4}$$

Notably, LLaDA employs a masking ratio that varies randomly between 0 and 1 while BERT [22] uses a fixed ratio. The subtle differences have significant implications, especially at scale: as shown in

---

[6]Here, we focus on the approach of LLaDA. A rigorous formulation of MDM is provided in Appendix A for interested readers.

Eq. (4), LLaDA is a principled generative model with the potential to perform in-context learning and instruction-following naturally, akin to LLMs. Moreover, its generative perspective implies strong scalability with large data and models as discussed in Sec. 1. In addition, MaskGIT [23] adopts a heuristic training objective, which misses the $\frac{1}{t}$ term compared to Eq. (3), and lacks a theoretical link to maximum likelihood. We emphasize that it is precisely the theoretical foundation of maximum likelihood estimation that motivated us to scale discrete diffusion models for language modeling.

## 2.2 Pre-training

LLaDA employs a Transformer [7] as the mask predictor, similar to existing LLMs. However, LLaDA does not use a causal mask, as its formulation allows it to see the entire input for predictions.

We trained two variants of LLaDA with different sizes: 1B and 8B. We summarize the model architecture of LLaDA 8B and LLaMA3 8B [6] here, and details are provided in Appendix B.2. We have ensured consistency in most hyperparameters while making several necessary modifications. We use vanilla multi-head attention instead of grouped query attention [24] for simplicity, as LLaDA is incompatible with KV caching, resulting in a different number of key and value heads. Consequently, the attention layer has more parameters, and we reduce the FFN dimension to maintain a comparable model size. Additionally, the vocabulary size differs due to a tokenizer [4] adapted on our data.

The LLaDA model is pre-trained on a dataset comprising 2.3 trillion (T) tokens, adhering to a data protocol that aligns closely with existing LLMs [25, 26], without the incorporation of any special techniques. The data are derived from online corpora, with low-quality content filtered through manually designed rules and LLM-based approaches. Beyond general text, the dataset encompasses high-quality code, math, and multilingual data. Please refer to Appendix B.1 for more details about datasets. The mixing of data sources and domains is guided by scaled-down ARMs. The pre-training process utilizes a fixed sequence length of 4096 tokens, incurring a total computational cost of 0.13 million H800 GPU hours, similar to ARMs of the same scale and dataset size.

For a training sequence $x_0$, we randomly sample $t \in [0, 1]$, mask each token independently with the same probability $t$ to obtain $x_t$ (see Fig. 2 (a)) and estimate Eq. (3) via the Monte Carlo method for stochastic gradient descent training. In addition, following Nie et al. [27], to enhance the ability of LLaDA to handle variable-length data, we set 1% of the pre-training data to a random length that is uniformly sampled from the range $[1, 4096]$.

We adopted the Warmup-Stable-Decay [28] learning rate scheduler to monitor the training progress without interrupting continuous training. Specifically, we linearly increased the learning rate from 0 to $4 \times 10^{-4}$ over the first 2000 iterations and maintained it at $4 \times 10^{-4}$. After processing 1.2T tokens, we decayed the learning rate to $1 \times 10^{-4}$ and held it constant for the next 0.8T tokens to ensure stable training. Finally, we linearly reduced the learning rate from $1 \times 10^{-4}$ to $1 \times 10^{-5}$ for the last 0.3T tokens. Furthermore, we utilized the AdamW optimizer [29] with a weight decay of 0.1, a batch size of 1280, and a local batch size of 4 per GPU. The 8B experiment was executed once, without any hyperparameter tuning.

## 2.3 Supervised Fine-Tuning

We enhance the capability of LLaDA to follow instructions by supervised fine-tuning (SFT) with paired data $(p_0, r_0)$, where $p_0$ is the prompt and $r_0$ denotes the response. This is the simplest and most basic post-training method for LLMs. Technically, this requires to model the conditional distribution $p_\theta(r_0|p_0)$ instead of $p_\theta(x_0)$ in pre-training.

The implementation is similar to pre-training. As shown in Fig. 2 (b), we leave the prompt unchanged and mask the tokens in the response independently, as done for $x_0$. Then, we feed both the prompt and the masked response $r_t$ to the pre-trained mask predictor to compute the loss for SFT:

$$-\mathbb{E}_{t,p_0,r_0,r_t} \left[ \frac{1}{t} \sum_{i=1}^{L'} \mathbf{1}[r_t^i = \mathbf{M}] \log p_\theta(r_0^i|p_0, r_t) \right], \tag{5}$$

where $L'$ denotes a dynamic length specified later, and all other notations remain the same as before.

Note that this approach is fully compatible with pre-training. Essentially, the concatenation of $p_0$ and $r_0$ can be treated as clean pre-training data $x_0$, while the concatenation of $p_0$ and $r_t$ serves as the

masked version $x_t$. The process is identical to pre-training, with the only difference being that all masked tokens happen to appear in the $r_0$ portion.

The LLaDA 8B model undergoes SFT on a dataset comprising 4.5 million pairs. Consistent with the pre-training process, both data preparation and training follow the SFT protocols utilized in existing LLMs [25, 26], without introducing any additional techniques to optimize LLaDA's performance. The dataset spans multiple domains, including code, mathematics, and instruction-following. We append |EOS| tokens to the end of short pairs in each mini-batch to ensure equal lengths across all data. We treat |EOS| as a normal token during training and remove it during sampling, enabling LLaDA to control the response length automatically. Please refer to Appendix B.1 for more details.

We train for 3 epochs on the SFT data using a similar schedule to the pre-training phase. The learning rate is linearly increased from 0 to $2.5 \times 10^{-5}$ over the first 50 iterations and then kept constant. During the final $10\%$ of iterations, it is linearly reduced to $2.5 \times 10^{-6}$. Additionally, we set the weight decay to $0.1$, the global batch size to $256$, and the local batch size to $2$ per GPU. The SFT experiment was executed once, without any hyperparameter tuning.

## 2.4 Inference

As a generative model, LLaDA can sample new text and evaluate the likelihood of candidate text *in a diffusion manner instead of the left-to-right autoregressive fashion*.

We begin with the reverse generation process. As illustrated in Fig. 2 (c), given a prompt $p_0$, we discretize the reverse process to sample from the model distribution $p_\theta(r_0|p_0)$, starting from a fully masked response. The total number of sampling steps is a hyperparameter, which naturally provides LLaDA with a trade-off between efficiency and sample quality, as analyzed in Sec. 3.3. We employ uniformly distributed timesteps by default. In addition, the generation length is also treated as a hyperparameter, specifying the length of the fully masked sentence at the beginning of the sampling process. After generation, tokens appearing after the |EOS| token are discarded. As detailed in Appendix B.5, since both pre-training and SFT are conducted using datasets with variable lengths, the final results are insensitive to this length hyperparameter.

At an intermediate step from time $t \in (0, 1]$ to $s \in [0, t)$, we feed both $p_0$ and $r_t$ into the mask predictor and predict all masked tokens simultaneously. Subsequently, we remask $\frac{s}{t}$ of the predicted tokens in expectation to obtain $r_s$, ensuring that the transition of the reverse process aligns with the forward process for accurate sampling [18–20]. In principle, the remasking strategy should be purely random. However, inspired by the annealing tricks of sampling in LLMs [4, 30], we adopt a low-confidence remasking strategy, where $\frac{s}{t}$ of predicted tokens with the lowest confidence are remarked based on the predictions, same as the approach of Chang et al. [23].

We mention that LLaDA enables flexible sampling. In particular, it supports autoregressive and block diffusion [31] sampling directly after the pre-training or SFT processes described above, without requiring any further modifications or training. We provide a detailed analysis in Appendix B.4. Nevertheless, the diffusion sampling (i.e., the reverse generation process) yields the best performance and is adopted as the default throughout this paper, especially for all experiments presented in Sec. 3.

For conditional likelihood evaluation, we can naturally utilize the upper bound in Eq. (5). However, we find that the following equivalent form [20] exhibits lower variance and is more stable:

$$-\mathbb{E}_{l,r_0,r_l}\left[\frac{L}{l}\sum_{i=1}^{L}\mathbf{1}[r_l^i = \mathrm{M}]\log p_\theta(r_0^i|p_0, r_l)\right], \tag{6}$$

where $L$ is the sequence length of $r_0$, $l$ is uniformly sampled from $\{1, 2, \ldots, L\}$, and $r_l$ is obtained by uniformly sampling $l$ tokens from $r_0$ without replacement for masking.

We present the training and inference algorithms, along with theoretical details, in Appendix A.

## 3 Experiments

We evaluate the scalability, instruction-following, and in-context learning capabilities of LLaDA on standard benchmarks, followed by analyses and case studies to provide a comprehensive assessment.

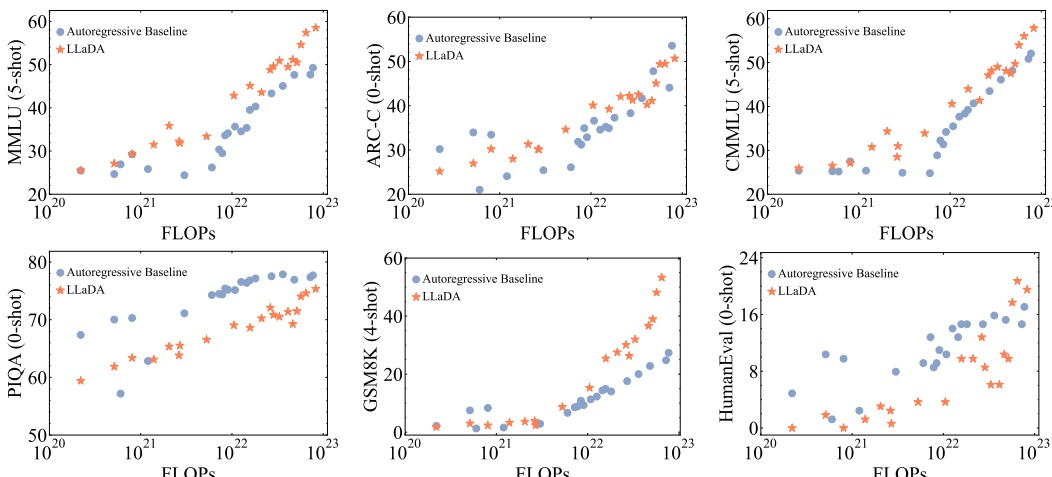

Figure 3: **Scalability of LLaDA.** We evaluate the performance of LLaDA and our ARM baselines trained on the same data across increasing pre-training computational FLOPs. LLaDA exhibits strong scalability, matching the overall performance of ARMs on six tasks.

## 3.1 Scalability of LLaDA on Language Tasks

We first investigate the *scalability* of LLaDA on downstream tasks in comparison with the ARM baselines we constructed. Specifically, at the 1B scale, we ensured that LLaDA and ARM shared the same architecture, data, and all other configurations. At larger scales, we also report results for LLaDA and ARM models of slightly different sizes trained on the same data due to resource limitations. Please refer to Appendix B.2 for more details. We use the pre-training computational cost as a unified scaling metric. For evaluation, we focused on six standard and diverse tasks.

Fig. 3 shows that LLaDA demonstrates impressive scalability, with its overall trend highly competitive with ARMs. Notably, on tasks such as MMLU and GSM8K, LLaDA exhibits even stronger scalability. Even on relatively weaker tasks like PIQA, the performance gap with ARMs narrows as scale increases. To account for the influence of outliers, we opted not to fit quantitative curves, avoiding potential misinterpretation. Nevertheless, the results clearly demonstrate the scalability of LLaDA.

Considering LLaDA's advantages on certain benchmarks, we hypothesize that this performance gain stems from a key architectural difference: while autoregressive models optimize only left-to-right conditional probabilities, LLaDA is trained to consider multiple conditioning directions, as detailed in Appendix A.2, which may offer greater flexibility and lead to better generalization. This hypothesis is motivated by LLaDA's strong performance on reversal reasoning in Sec. 3.3 and the ablation studies on sampling strategies in Appendix B.4.

Nie et al. [27] suggests that MDM requires 16 times more computation than ARM to achieve the same likelihood. However, key differences make our findings more broadly applicable. In particular, likelihood is a relatively indirect metric for downstream task performance, and diffusion optimizes a bound of the likelihood, making it not directly comparable to ARM. Additionally, we extended the scaling range from $10^{18} \sim 10^{20}$ FLOPs in Nie et al. [27] to $10^{20} \sim 10^{23}$ FLOPs in this work.

## 3.2 Benchmark Results

To comprehensively evaluate the *in-context learning* and *instruction-following* capabilities of LLaDA 8B, we conducted detailed comparisons with existing LLMs [6, 21, 25, 26, 32, 33] of similar scale. Task selection and evaluation protocols followed existing studies, covering popular benchmarks in general tasks, mathematics, code, and Chinese. Further details are provided in Appendix B.6. For a more direct comparison, we re-evaluated representative LLMs [6, 21] in our implementation.

As shown in Tab. 1, after pretraining on 2.3T tokens, LLaDA 8B Base demonstrates remarkable performance, surpassing LLaMA2 7B Base on nearly all tasks, and is overall competitive with LLaMA3 8B Base. LLaDA shows advantages in math and Chinese tasks. We conjecture that the

Table 1: **Benchmark Results of Pre-trained LLMs.** [*] indicates that models are evaluated under the same protocol, detailed in Appendix B.6. Results indicated by [†] and [¶] are sourced from Yang et al. [25, 26] and Bi et al. [32] respectively. The numbers in parentheses represent the number of shots used for in-context learning. "-" indicates unknown data.

| | LLaDA 8B[*] | LLaMA3 8B[*] | LLaMA2 7B[*] | Qwen2 7B[†] | Qwen2.5 7B[†] | Mistral 7B[†] | Deepseek 7B[¶] |
|---|---|---|---|---|---|---|---|
| Model | Diffusion | AR | AR | AR | AR | AR | AR |
| Training tokens | 2.3T | 15T | 2T | 7T | 18T | - | 2T |
| *General Tasks* | | | | | | | |
| MMLU | **65.9** (5) | 65.4 (5) | 45.9 (5) | 70.3 (5) | 74.2 (5) | 64.2 (5) | 48.2 (5) |
| BBH | 49.7 (3) | **62.1** (3) | 39.4 (3) | 62.3 (3) | 70.4 (3) | 56.1 (3) | 39.5 (3) |
| ARC-C | 45.9 (0) | **53.1** (0) | 46.3 (0) | 60.6 (25) | 63.7 (25) | 60.0 (25) | 48.1 (0) |
| Hellaswag | 70.5 (0) | **79.1** (0) | 76.0 (0) | 80.7 (10) | 80.2 (10) | 83.3 (10) | 75.4 (0) |
| TruthfulQA | **46.1** (0) | 44.0 (0) | 39.0 (0) | 54.2 (0) | 56.4 (0) | 42.2 (0) | - |
| WinoGrande | 74.8 (5) | **77.3** (5) | 72.5 (5) | 77.0 (5) | 75.9 (5) | 78.4 (5) | 70.5 (0) |
| PIQA | 73.6 (0) | **80.6** (0) | 79.1 (0) | - | - | - | 79.2 (0) |
| *Mathematics & Science* | | | | | | | |
| GSM8K | **70.3** (4) | 48.7 (4) | 13.1 (4) | 80.2 (4) | 85.4 (4) | 36.2 (4) | 17.4 (8) |
| Math | **31.4** (4) | 16.0 (4) | 4.3 (4) | 43.5 (4) | 49.8 (4) | 10.2 (4) | 6.0 (4) |
| GPQA | 25.2 (5) | **25.9** (5) | 25.7 (5) | 30.8 (5) | 36.4 (5) | 24.7 (5) | - |
| *Code* | | | | | | | |
| HumanEval | **35.4** (0) | 34.8 (0) | 12.8 (0) | 51.2 (0) | 57.9 (0) | 29.3 (0) | 26.2 (0) |
| HumanEval-FIM | **73.8** (2) | 73.3 (2) | 26.9 (2) | - | - | - | - |
| MBPP | 40.0 (4) | **48.8** (4) | 23.2 (4) | 64.2 (0) | 74.9 (0) | 51.1 (0) | 39.0 (3) |
| *Chinese* | | | | | | | |
| CMMLU | **69.9** (5) | 50.7 (5) | 32.5 (5) | 83.9 (5) | - | - | 47.2 (5) |
| C-Eval | **70.5** (5) | 51.7 (5) | 34.0 (5) | 83.2 (5) | - | - | 45.0 (5) |

Table 2: **Benchmark Results of Post-trained LLMs.** LLaDA only employs an SFT procedure, while other models have extra reinforcement learning (RL) alignment. [*] indicates models are evaluated under the same protocol, detailed in Appendix B.6. Results indicated by [†] and [¶] are sourced from Yang et al. [26] and Bi et al. [32] respectively. The numbers in parentheses represent the number of shots used for in-context learning. "-" indicates unknown data.

| | LLaDA 8B[*] | LLaMA3 8B[*] | LLaMA2 7B[*] | Qwen2 7B[†] | Qwen2.5 7B[†] | Gemma2 9B[†] | Deepseek 7B[¶] |
|---|---|---|---|---|---|---|---|
| Model | Diffusion | AR | AR | AR | AR | AR | AR |
| Training tokens | 2.3T | 15T | 2T | 7T | 18T | 8T | 2T |
| Post-training | SFT | SFT+RL | SFT+RL | SFT+RL | SFT+RL | SFT+RL | SFT+RL |
| Alignment pairs | 4.5M | - | - | 0.5M + - | 1M + 0.15M | - | 1.5M + - |
| *General Tasks* | | | | | | | |
| MMLU | 65.5 (5) | **68.4** (5) | 44.1 (5) | - | - | - | 49.4 (0) |
| MMLU-pro | 37.0 (0) | **41.9** (0) | 4.6 (0) | 44.1 (5) | 56.3 (5) | 52.1 (5) | - |
| Hellaswag | 74.6 (0) | **75.5** (0) | 51.5 (0) | - | - | - | 68.5 (-) |
| ARC-C | **88.5** (0) | 82.4 (0) | 57.3 (0) | - | - | - | 49.4 (-) |
| *Mathematics & Science* | | | | | | | |
| GSM8K | 69.4 (4) | **78.3** (4) | 29.0 (4) | 85.7 (0) | 91.6 (0) | 76.7 (0) | 63.0 (0) |
| Math | **31.9** (0) | 29.6 (0) | 3.8 (0) | 52.9 (0) | 75.5 (0) | 44.3 (0) | 15.8 (0) |
| GPQA | **33.3** (5) | 31.9 (5) | 28.4 (5) | 34.3 (0) | 36.4 (0) | 32.8 (0) | - |
| *Code* | | | | | | | |
| HumanEval | 49.4 (0) | **59.8** (0) | 16.5 (0) | 79.9 (0) | 84.8 (0) | 68.9 (0) | 48.2 (-) |
| MBPP | 41.0 (4) | **57.6** (4) | 20.6 (4) | 67.2 (0) | 79.2 (0) | 74.9 (0) | 35.2 (-) |

strengths stem from the same factors as its relatively weaker performance in some tasks—differences in data quality and distribution, largely due to the closed-source situation of LLM datasets.

Notably, we have carefully ruled out the possibility of data leakage by taking GSM8K as an example. First, as shown in Fig. 3, LLaDA outperformed ARM baselines regarding GSM8K. Moreover, the conclusion remains on a fully unseen GSM8K-like task [34] in Appendix B.8.

Further, Tab. 2 compares the performance of LLaDA 8B Instruct with existing LLMs. SFT improved LLaDA's performance on most downstream tasks. A few metrics, such as MMLU, showed declines, possibly due to the suboptimal quality of the SFT data. Overall, since we did not perform alignment

Table 3: **Visualization of the Sampling Process and a Generated Multi-round Dialogue.** In the response of LLaDA, darker colors indicate tokens predicted in the later stages of sampling, while lighter colors correspond to earlier predictions.

---

*Sampling Process*

| User | Lily can run 12 kilometers per hour for 4 hours. After that, she runs 6 kilometers per hour. How many kilometers can she run in 8 hours? |
|---|---|
| LLaDA | In 4 hours , Lily runs 12 * 4 = 48 kilometers . After 4 hours she runs 6 * 4 = 24 kilometers . In total she runs 48 + 24 = 72 kilometers in 8 hours . Result 72 |

---

*Multi-round Dialogue Case*

| User | Could you please share with me what the initial two lines of the renowned poem 'The Road Not Taken'? |
|---|---|
| LLaDA | Certainly! The initial two lines of the renowned poem "The Road Not Taken" by Robert Frost are: "Two roads diverged in a yellow wood, And sorry I could not travel both." These lines set the stage for the poem, introducing the idea of a speaker facing a choice between two paths. |
| User | Please help me translate into Chinese. |
| LLaDA | Sure! The first two lines of "The Road Not Taken" by Robert Frost can be translated into Chinese as: "两条路分岔在黄色的树林中，遗憾我不能同时走" |
| User | Please translate into German. |
| LLaDA | Sure! The first two lines of "The Road Not Taken" by Robert Frost can be translated into German as: "Zwei Wege trennten sich im gelben Wald, und leider konnte ich nicht beide tun." |
| User | Please also write a poem about life choices. I require 5 sentences, and the first word of each sentence starts with C as the first letter. |
| LLaDA | Certainly! Here's a poem about life choices, with each sentence starting with the letter "C": Choices cascade through life's journey, Creating connections with each decision, Careful consideration leads to better paths, Courageous choices shape our destiny, Choosing wisely guides us through the years. |

---

with reinforcement learning (RL), our results are slightly behind LLaMA3 8B Instruct, though the gaps in many metrics remain small. Notably, even with only SFT, LLaDA demonstrates impressive instruction-following abilities, as detailed in Sec. 3.4. We leave RL-based alignment for future work.

All results in Sec. 3 are based on pure diffusion methods, as they achieve better overall performance than approaches incorporating autoregressive components. Specifically, we use Eq. (6) for conditional likelihood estimation and apply low-confidence remasking for sampling. For LLaDA 8B Instruct, block diffusion style sampling performs better on GSM8K and Math, with scores of 78.6 and 42.2, compared to 69.4 and 31.9 in Tab. 2. This gain is due to extensive |EOS| token padding in the SFT data, causing early termination in low-confidence remasking. Please refer to Appendix B.4 for details.

Overall, despite the lack of data transparency, we have made every effort to adopt standardized procedures and introduce diverse tasks, we believe they sufficiently demonstrate the extraordinary capabilities of LLaDA, which is the only competitive non-autoregressive model to our knowledge.

## 3.3 Reversal Reasoning and Analyses

To quantify the reversal reasoning [15] ability of models, we follow the protocol established in Allen-Zhu and Li [35]. Specifically, we construct a dataset of 496 famous Chinese poem sentence pairs. Given a sentence from a poem, models are tasked with generating the subsequent line (forward) or the preceding line (reversal) without additional fine-tuning. Examples can be found in Appendix B.9. This setting provides a straightforward and more realistic evaluation compared to previous studies [27, 36].

As shown in Tab. 4, LLaDA effectively addresses the *reversal curse* [15], demonstrating consistent zero-shot performance across both forward and reversal tasks. In contrast, both Qwen 2.5 and GPT-4o exhibit a significant gap between the two. The results on forward generation confirm that both ARMs are strong, benefiting from significantly larger datasets and greater computational resources than LLaDA. However, LLaDA outperforms both by a large margin in the reversal task.

We did not design anything special for reversal tasks. Intuitively, LLaDA treats tokens uniformly without inductive bias, leading to balanced performance. See Appendix A.2 for details.

We also analyze the effect of different sampling strategies for LLaDA, including autoregressive sampling, block diffusion [31] sampling, and pure diffusion sampling, showing that pure diffusion sampling achieves the best overall performance, as detailed in Appendix B.4.

Table 4: **Comparison on the Poem Completion task.**

|  | Forward | Reversal |
|---|---|---|
| GPT-4o (2024-08-06) | **82.7** | 34.3 |
| Qwen2.5-7B Instruct | 75.9 | 38.0 |
| LLaDA-8B Instruct | 51.8 | **45.6** |

In addition, we examine LLaDA's sampling speed and memory consumption, showing that it enables a flexible trade-off between generation quality and speed. See Appendix B.7 for more details.

Classifier-free guidance (CFG) [37, 27] is a widely used technique in diffusion models to improve generation quality. To ensure a fair comparison with ARMs, we do not apply CFG to LLaDA in the main text. However, we show that LLaDA is compatible with CFG and consistently benefits from its application. See Appendix B.3 for more details.

### 3.4 Case Studies

We present samples generated by LLaDA 8B Instruct in Tab. 3, showcasing its instruction-following capabilities. First, the table illustrates LLaDA's ability to generate coherent, fluent, and extended text in a non-autoregressive manner. Second, it highlights the model's multi-turn dialogue capability, effectively retaining conversation history and producing contextually appropriate responses across multiple languages. Such *chat* capabilities of LLaDA are impressive, as it departs from conventional ARMs for the first time, to the best of our knowledge. See more case studies in Appendix B.10.

## 4 Related Work

Diffusion models [38–40] have achieved remarkable success in visual domains but remain unverified for large-scale (e.g., models trained with over $10^{23}$ FLOPs) language modeling, despite growing interest and extensive research efforts.

A simple approach is to continuousize text data and apply continuous diffusion models directly [41–51]. Alternatively, some methods model continuous parameters of discrete distributions instead [52–56]. However, scalability remains a significant challenge for these approaches. For instance, a 1B model may require 64 times the compute of an ARM to achieve comparable performance [57].

Another approach replaces continuous diffusion with discrete processes featuring new forward and reverse dynamics, leading to numerous variants [58–71]. The original diffusion model paper [38] introduced both continuous-state and discrete-state transition kernels under a unified diffusion framework. Austin et al. [16] was among the pioneering works that introduced discrete diffusion models into language modeling, demonstrating the feasibility of this approach. Lou et al. [17] showed that masked diffusion, as a special case of discrete diffusion, achieves perplexity comparable to or surpassing ARMs at GPT-2 scale. Shi et al. [18], Sahoo et al. [19], Ou et al. [20] established fundamental theoretical results, which motivated our model design, training, and inference (see Appendix A for details). Nie et al. [27] introduced the scaling laws for MDMs in language modeling and explored how MDMs can be leveraged for language tasks such as question answering at the GPT-2 scale. Gong et al. [72] demonstrated the potential of fine-tuning an ARM within the MDM framework. However, the improvements observed by Gong et al. [72] are limited to specific metrics, and their approach does not address the performance achievable through pure diffusion-based training. Concurrent work [73] demonstrates the potential of diffusion language models in code generation and highlights their advantages in inference efficiency. Nonetheless, as it is a closed-source product, specific details such as training procedures and sampling methods remain unknown.

In comparison, this study scales MDM to an unprecedented size of 8B parameters from scratch, achieving performance comparable to leading LLMs such as LLaMA 3.

Additionally, a parallel line of work on image generation [23, 74, 75] aligns well with the application of MDMs to text data. Moreover, MDMs have also shown promise in other domains such as protein

generation [76, 77], where they have achieved promising results. Notably, a series of studies [31, 78–87] have explored techniques such as architectural optimization, distillation, and sampling algorithm design to accelerate MDMs sampling.

## 5 Conclusion and Discussion

We introduce LLaDA, a diffusion language model trained from scratch with an unprecedented scale of 8B parameters. LLaDA demonstrates strong capabilities in scalability, in-context learning, and instruction-following, achieving performance comparable to strong LLMs such as LLaMA3. In addition, LLaDA offers unique advantages, such as bidirectional modeling and enhanced robustness, effectively addressing the relevant limitations of existing LLMs. Our findings show the promise of diffusion models for language modeling at scale and challenge the common assumption that these essential capabilities are inherently tied to ARMs. These results represent a new paradigm for language modeling and uncover novel insights, demonstrating a high degree of scientific innovation.

**Limitations.** While promising, the full potential of diffusion models remains to be fully explored. Several limitations of this work present significant opportunities for future research. The generation length is a user-specified hyperparameter. Although LLaDA is insensitive to this hyperparameter as detailed in Appendix B.5, we believe that adopting an adaptive generation length would offer a more efficient solution. Due to computational constraints, direct comparisons between LLaDA and ARMs—such as training on identical datasets—were restricted to a computational budget of less than $10^{23}$ FLOPs. To allocate resources for training the largest possible LLaDA model and showcasing its potential, we were unable to scale the ARM baseline to the same extent. Moreover, no specialized attention mechanisms or position embeddings were designed for LLaDA, nor were any system-level architectural optimizations such as KV cache applied. On the inference side, more efficient and controllable [37, 88, 89] sampling algorithms remain preliminary. Furthermore, LLaDA has yet to undergo alignment with reinforcement learning [90, 91], which is crucial for improving its performance and alignment with human intent.

Looking ahead, both the model scale and the amount of training data for LLaDA remain smaller than those of leading ARM counterparts [6, 26, 92–95], highlighting the need for further scaling to fully evaluate its capabilities. In addition, LLaDA's ability to process multi-modal data remains unexplored. Its impact on prompt tuning techniques [96] and integration into agent-based systems [97, 98] is still not fully understood. Finally, a systematic investigation into post-training for LLaDA (e.g., O1-like systems [99, 100]) is needed to further unlock the potential of diffusion language models.

## Acknowledgements

This work was supported by the National Natural Science Foundation of China (No. 92470118); Beijing Natural Science Foundation (No. L247030); Beijing Nova Program (No. 20220484044); the Beijing Major Science and Technology Project under Contract no. Z251100008425002; the fund for building world-class universities (disciplines) of Renmin University of China; and Ant Group Research Fund.

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

**Algorithm 1** Pre-training of LLaDA

---

**Require:** mask predictor $p_\theta$, data distribution $p_{\text{data}}$
1: **repeat**
2:   $x_0 \sim p_{\text{data}}$          # with a probability of 1%, the sequence length of $x_0$ follows U$[1, 4096]$
3:   $t \sim$ U$(0, 1]$
4:   $x_t \sim q_{t|0}(x_t|x_0)$                                    # $q_{t|0}$ is defined in Eq. (7)
5:   Calculate $\mathcal{L} = -\frac{1}{t*L}\sum_{i=1}^{L} \mathbf{1}[x_t^i = \text{M}] \log p_\theta(x_0^i|x_t)$        # $L$ is the sequence length of $x_0$
6:   Calculate $\nabla_\theta \mathcal{L}$ and run optimizer.
7: **until** Converged
8: **Return** $p_\theta$

---

---

**Algorithm 2** Supervised Fine-Tuning of LLaDA

---

**Require:** mask predictor $p_\theta$, pair data distribution $p_{\text{data}}$
1: **repeat**
2:   $p_0, r_0 \sim p_{\text{data}}$               # please refer to Appendix B.1 for details about the SFT dat
3:   $t \sim$ U$(0, 1]$
4:   $r_t \sim q_{t|0}(r_t|r_0)$                                    # $q_{t|0}$ is defined in Eq. (7)
5:   Calculate $\mathcal{L} = -\frac{1}{t*L'}\sum_{i=1}^{L'} \mathbf{1}[r_t^i = \text{M}] \log p_\theta(r_0^i|p_0, r_t)$     # $L'$ is the sequence length of $r_0$
6:   Calculate $\nabla_\theta \mathcal{L}$ and run optimizer.
7: **until** Converged
8: **Return** $p_\theta$

---

---

**Algorithm 3** Conditional Log-likelihood Evaluation of LLaDA

---

**Require:** mask predictor $p_\theta$, prompt $p_0$, response $r_0$, the number of Monte Carlo estimations $n_{mc}$
1: log_likelihood $= 0$
2: **for** $i \leftarrow 1$ to $n_{mc}$ **do**
3:   $l \sim \{1, 2, \ldots, L\}$                                    # $L$ is the sequence length of $r_0$
4:   Obtain $r_l$ by uniformly sampling $l$ tokens from $r_0$ without replacement for masking
5:   log_likelihood = log_likelihood $+ \frac{L}{l}\sum_{i=1}^{L} \mathbf{1}[r_l^i = \text{M}] \log p_\theta(r_0^i|p_0, r_l)$
6: **end for**
7: log_likelihood = log_likelihood$/n_{mc}$
8: **Return** log_likelihood

---

# A   Formulation of Masked Diffusion Models

## A.1   Training

MDMs [16–20] define the model distribution $p_\theta(x_0)$ in a manner distinct from autoregressive models.

These models introduce a forward process $\{x_t\}$ indexed by a time $t \in [0, 1]$. This process gradually and independently masks all tokens in the sequence $x_0$. At time $t = 0$, the data point $x_0$ is fully observed with no masks, while for $t \in (0, 1]$, $x_t$ represents latent variables with varying mask ratios in expectation.

Formally, the conditional distribution of $x_t$ given $x_0$ is defined by a fully factorized form:

$$q_{t|0}(x_t|x_0) = \prod_{i=1}^{L} q_{t|0}(x_t^i|x_0^i), \tag{7}$$

where the conditional distribution for each token is given by:

$$q_{t|0}(x_t^i|x_0^i) = \begin{cases} 1 - t, & x_t^i = x_0^i, \\ t, & x_t^i = \text{M}. \end{cases} \tag{8}$$

Here, M denotes the mask token. Intuitively, each token either remains unchanged or is masked, with the probability of being masked increasing linearly as $t$ progresses from 0 to 1. At $t = 1$, all

---

**Algorithm 4** Random Remasking Strategy of LLaDA

---

**Require:** mask predictor $p_\theta$, prompt $p_0$, answer length $L$, sampling steps $N$
1: Set $r_1$ is a fully masked sequence of length $L$.
2: **for** $t \leftarrow 1$ **down to** $\frac{1}{N}$ **step** $\frac{1}{N}$ **do**
3:      $s = t - \frac{1}{N}$
4:      $r_0 = \arg\max_{r_0} p_\theta(r_0|p_0, r_t)$ # we employ greedy sampling when predicting masked tokens
5:      **for** $i \leftarrow 1$ to $L$ **do**
6:         **if** $r_t^i \neq M$ **then**
7:            $r_0^i = r_t^i$
8:         **else**
9:            with probability $\frac{s}{t}$, $r_0^i$ is set to $M$
10:        **end if**
11:      **end for**
12:      $r_s = r_0$
13: **end for**
14: **Return** $r_0$

---

tokens are guaranteed to be masked, meaning that $x_1$ follows a Dirac distribution concentrated on a sequence of fully masked tokens. Notably, the linear masking probability is analogous to but distinct from, the noise schedule in continuous diffusion models [38–40]. This linearity is motivated by the assumption that the information in the text is proportional to the number of tokens on average, making it reasonable to lose information linearly during the forward process.

The forward process is not only reversible but also corresponds to a reverse process that is fully factorized across all tokens. The reverse process, from time $t = 1$ to $0$, generates new data from sequences of fully masked tokens. The conditional distribution for the reverse process, for $0 \leq s < t \leq 1$, is factorized as:

$$q_{s|t}(x_s|x_t) = \prod_{i=1}^{L} q_{s|t}(x_s^i|x_t), \tag{9}$$

where the conditional distribution for each token is:

$$q_{s|t}(x_s^i|x_t) = \begin{cases} 1, & x_t^i \neq M, \, x_s^i = x_t^i, \\ \frac{s}{t}, & x_t^i = M, \, x_s^i = M, \\ \frac{t-s}{t} q_{0|t}(x_s^i|x_t), & x_t^i = M, \, x_s^i \neq M, \\ 0, & \text{otherwise.} \end{cases} \tag{10}$$

Thus, the key function to estimate is the conditional distribution $q_{0|t}(x_s^i|x_t)$, which predicts the original token if it is masked in the input $x_t$. This is analogous to the *data prediction* form in continuous diffusion models.

As proven in [20], an equivalent yet *time-free* parameterization can be derived as:

$$q_{0|t}(x_s^i|x_t) = p_{\text{data}}(x_0^i|x_t^{\text{UM}}), \quad \forall i \text{ such that } x_t^i = M, \tag{11}$$

where $x_t^{\text{UM}}$ denotes the collection of unmasked tokens in $x_t$, which is identical to the corresponding tokens in the original data $x_0$ since unmasked tokens are solely determined by $x_0$ and are independent of time $t$. Intuitively, this implies that estimating the data prediction function is equivalent to estimating the conditional distributions on clean data, which is time-invariant. Consequently, the time $t$ need not be provided as input to the parametric model.

Although the development of masked diffusion is nontrivial, the implementation is straightforward. We first introduce the *mask predictor*, a parametric model $p_\theta(\cdot|x_t)$ (e.g., a Transformer without causal mask), which takes $x_t$ for any $t$ as input and predict all masked tokens simultaneously. Then, we define the model distribution $p_\theta(x_0)$ as follows: starting with $x_1$ as a sequence of fully masked tokens, we simulate an approximate reverse process parameterized by $p_\theta(\cdot|x_t)$ from $t = 1$ to $0$. The marginal distribution induced at $t = 0$ then represents the model distribution $p_\theta(x_0)$.

**Algorithm 5** Low-confidence Remasking Strategy of LLaDA

---

**Require:** mask predictor $p_\theta$, prompt $p_0$, answer length $L$, sampling steps $N$
1: Set $r_1$ is a fully masked sequence of length $L$.
2: **for** $t \leftarrow 1$ **down to** $\frac{1}{N}$ **step** $\frac{1}{N}$ **do**
3:     $s = t - \frac{1}{N}$
4:     **for** $i \leftarrow 1$ to $L$ **do**
5:       **if** $r_t^i \neq M$ **then**
6:         $r_0^i = r_t^i, c^i = 1$
7:       **else**
8:         $r_0^i = \arg\max_{r_0^i} p_\theta(r_0^i|p_0, r_t)$
9:         $c^i = p_\theta(r_0^i|p_0, r_t)_{r_0^i}$
10:       **end if**
11:     **end for**
12:     $n_{un} = \lfloor L(1-s) \rfloor$              # the number of unmasked tokens is $n_{un}$ in timestep $s$
13:     **for** $i \leftarrow 1$ to $L$ **do**
14:       **if** $c^i \in \text{Lowest} - n_{un}\left(\{c^i\}_1^L\right)$ **then**
15:         $r_0^i = M$       # the $n_{un}$ positions with the least confidence are selected for remasking.
16:       **end if**
17:     **end for**
18:     $r_s = r_0$
19: **end for**
20: **Return** $r_0$

---

Formally, the mask predictor is trained using a cross-entropy loss with masking:

$$\mathcal{L}(\theta) \triangleq -\mathbb{E}_{t,x_0,x_t}\left[\frac{1}{t}\sum_{i=1}^{L}\mathbf{1}[x_t^i = M]\log p_\theta(x_0^i|x_t)\right], \tag{12}$$

where $x_0$ is sampled from the training data, $t$ is sampled uniformly from $[0, 1]$, and $x_t$ is sampled from $q_{t|0}(x_t|x_0)$. The indicator function $\mathbf{1}[\cdot]$ ensures that the cross-entropy loss is computed only for masked tokens. In Shi et al. [18], Sahoo et al. [19], Ou et al. [20], it has been proven that the loss function $\mathcal{L}(\theta)$ is an upper bound on the negative log-likelihood of the model distribution:

$$-\mathbb{E}_{x_0 \sim p_{\text{data}(x_0)}}\left[\log p_\theta(x_0)\right] \leq \mathcal{L}(\theta). \tag{13}$$

In summary, this principled approach trains a generative model by progressively masking tokens during a forward process and learning to recover the data distribution during a reverse process, all under the (approximate) maximum likelihood estimation framework.

## A.2 Inference

The cross-entropy loss in Eq. (12) has several equivalent forms [20]. The first one is given by

$$-\mathbb{E}_{l\sim\{1,2,\ldots,L\},x_0,x_l}\left[\frac{L}{l}\sum_{i=1}^{L}\mathbf{1}[x_l^i = M]\log p_\theta(x_0^i|x_l)\right], \tag{14}$$

where $l$ is uniformly sampled from $\{1, 2, \ldots, L\}$, and $x_l$ is obtained by uniformly sampling $l$ tokens from $x_0$ without replacement for masking. Despite masking exactly $l$ tokens is different from masking each token independently with probability $t$, these two masking methods lead to equivalent results in expectation [20].

While Eq. (12) and Eq. (14) share the same expectation, their variances differ. Intuitively, in Eq. (12), we expect $x_t$ to have a fraction of $t$ tokens masked. However, the randomness of the forward process (i.e., Eq. (7)) often causes deviations, especially when $x_t$ contains few tokens. In contrast, in Eq. (14), the fraction of masked tokens in $x_l$ is deterministically $\frac{l}{L}$. While a theoretical analysis depends on the data distribution, empirical results show that Eq. (12) requires over 1000 Monte Carlo estimates for stable results, whereas Eq. (14) achieves stability with only 128 estimates. In addition, we can simply modify Eq. (14) to its conditional version (i.e., Eq. (6)) based on Eq. (5).

Any-order autoregressive models (AO-ARM) [59, 101, 102] characterize the joint distribution autoregressively for all possible orders $\pi$ of the $L$ variables. To learn such a distribution, an AO-ARM utilizes a weight-sharing neural network to model all univariate conditionals and employs mask tokens to represent absent variables. During training, the expected negative log-likelihood over the uniform distribution of all orders $U_\pi$ is minimized:

$$-\mathbb{E}_{x_0, \pi \sim U_\pi} \left[ \sum_{i=1}^{L} \log p_\theta(x_0^{\pi(i)} | x_0^{\pi(<i)}; \pi) \right]. \tag{15}$$

Intuitively, $x_0^{\pi(<i)}$ can be understood as a masked token $x_t$ with index in $\pi(\geq i)$ being masked. It can be further proved that Eq. (15) is equivalent to Eq. (12). This connection explains the bidirectional reasoning capabilities of LLaDA, even though it was never used explicitly in the inference procedure.

In addition, Nie et al. [27] introduces unsupervised classifier-free guidance (CFG), a plug-and-play technique that balances alignment with prompts and text diversity. Specifically, unsupervised CFG employs the following modified mask predictor for inference:

$$\tilde{p}_\theta(r_0 | p_0, r_t) \propto \frac{p_\theta(r_0 | p_0, r_t)^{1+w}}{p_\theta(r_0 | m, r_t)^w}, \tag{16}$$

where $m$ is a mask sequence of the same length as $p_0$ and $w$ is a tunable hyperparameter that controls the strength of $p_0$. To ensure a fair comparison with ARMs, we do not apply CFG to LLaDA in the main text. However, as demonstrated in Appendix B.3, LLaDA is fully compatible with CFG and consistently exhibits improved performance when it is applied.

### A.3 Algorithms

In this section, we present the training and inference algorithms. Specifically, we introduce the pre-training and supervised fine-tuning algorithms in Algorithm 1 and Algorithm 2, respectively. In addition, the likelihood evaluation algorithm is provided in Algorithm 3. Finally, we present the reverse generation process in Algorithm 4 and Algorithm 5, which correspond to the random remasking and the low-confidence [23] remasking strategy, respectively.

## B   Experiments

### B.1   Data Collection and Preprocessing

In this section, we first introduce the data collection and filtering processes for both pre-training and SFT. We then describe how LLaDA leverages these datasets during training.

Our pre-training corpus is constructed from diverse publicly available sources, including web data, books, academic papers, social media, encyclopedias, mathematics, and code, with approximately 11% Chinese, 61% English, and 28% code. The data cleaning process involves PDF text extraction, deduplication, and harmful content filtering. To further ensure quality, we fine-tune a BERT [22] model for automated data quality annotation, enabling the selection of higher-quality samples. Our SFT dataset consists of 1 million human-annotated samples and 3.5 million synthetic samples, generated using methods similar to those proposed in Xu et al. [103], Wei et al. [104].

We concatenate the collected documents in the pre-training corpus and segment the text into fixed-length sequences according to the predefined sequence length.

For SFT, a dynamic sequence length strategy is employed, where |EOS| tokens are appended to the end of shorter pairs to ensure uniform sequence lengths across all samples within each mini-batch. Notably, the padding |EOS| tokens are treated as part of the response, i.e., masked and included in the training objective. The |EOS| tokens are removed from the generated outputs during sampling. This strategy ensures that the model learns to control the length of its responses by generating |EOS|, enabling the response length to align effectively with the given prompt.

In addition, for $n$-turn dialogues $(p_0^0, r_0^0, p_0^1, r_0^1, \ldots, p_0^{n-1}, r_0^{n-1})$, we treat it as $n$ single-turn dialogue pairs, i.e., $(p_0^0, r_0^0), (p_0^0 r_0^0 p_0^1, r_0^1), \ldots, (p_0^0 r_0^0 p_0^1 r_0^1 \ldots p_0^{n-1}, r_0^{n-1})$ and randomly sample one. This data partitioning strategy not only equips LLaDA with multi-turn dialogue capabilities but also aligns with the above |EOS| padding strategy.

Table 5: **Model Architecture.** We report the architectural configurations for our 1B and 7B ARM baselines, the 1B and 8B LLaDA models, and the 8B LLaMA3 model.

| | Our ARM Baseline 1B | LLaDA 1B | Our ARM Baseline 7B | LLaDA 8B | LLaMA3 8B |
|---|---|---|---|---|---|
| Layers | 22 | 22 | 28 | 32 | 32 |
| Model dimension | 2048 | 2048 | 4096 | 4096 | 4096 |
| Attention heads | 32 | 32 | 32 | 32 | 32 |
| Vocabulary size | 126,464 | 126,464 | 126,464 | 126,464 | 128,000 |
| FFN dimension | 5634 | 5634 | 13,440 | 12,288 | 14,336 |
| Key/Value heads | 4 | 4 | 8 | 32 | 8 |
| Total parameters | 1.49 B | 1.49 B | 6.83 B | 8.02 B | 8.03 B |
| Non-embedding parameters | 0.97 B | 0.97 B | 5.80 B | 6.98 B | 6.98 B |

Table 6: **Ablation on CFG.** CFG consistently improves the performance of LLaDA.

| | ARC-C | Hellaswag | TruthfulQA | WinoGrande | GPQA | PIQA |
|---|---|---|---|---|---|---|
| w/o CFG | 45.9 | 70.5 | 46.1 | **74.8** | 25.2 | 73.6 |
| w/ CFG | **47.9** | **72.5** | **46.4** | **74.8** | **26.1** | **74.4** |

## B.2 Details about Model Training

This section provides the training details of LLaDA and the corresponding ARM baselines.

Firstly, for efficiency, we trained an ARM and an MDM, both with 1.5B parameters and identical architectures. Additionally, we scaled the MDM to 8B parameters. Due to computational resource constraints, we did not train an 8B autoregressive model with the same architecture. Instead, we utilized our previously trained 7B autoregressive model for comparison. These four models are utilized in the scalability analysis in Sec. 3.1.

We adopted a Transformer architecture similar to LLaMA [6, 21] for the ARMs and MDMs we trained. Specifically, we employ RMSNorm [105] to stabilize training, use SwiGLU [106] as the activation function to enhance non-linearity, and integrate RoPE [107] for more expressive positional encoding. Tab. 5 provides an overview of the model architectures.

For the 1B and 7B ARM baselines, as well as the 1B and 8B LLaDA models, we utilized the AdamW optimizer [29] with a weight decay of 0.1 and adopted the Warmup-Stable-Decay [28] learning rate scheduler. The learning rate was linearly increased from 0 to the maximum value over the first 2000 iterations and then held constant. For LLaDA 8B, to ensure stable training, the learning rate was reduced once during pre-training, as detailed in Sec. 2.2. For the 1B ARM baseline and both the 1B and 8B LLaDA models, the maximum learning rate is set to $4 \times 10^{-4}$ with a batch size of 1280, without any hyperparameter tuning. For the 7B ARM baseline, the maximum learning rate is set to $4.2 \times 10^{-4}$ with a batch size of 4224, both selected via grid search.

Additionally, we employ the widely used $6ND$ formulation [108, 109] to calculate the training FLOPs in Fig. 3, where $N$ represents the number of non-embedding parameters, and $D$ denotes the total number of training tokens. The detailed results corresponding to Fig. 3 are provided in Tab. 18 and Tab. 19.

## B.3 Ablation on Classifier-free Guidance

This section presents an ablation study on classifier-free guidance (CFG). Theoretical details about CFG can be found in the Appendix A.2.

For simplicity, we select six representative benchmarks, including ARC-C, HellaSwag, TruthfulQA, WinoGrande, PIQA, and GPQA, and conduct experiments using LLaDA 8B Base. We search the CFG scale in $\{0.5, 1, 1.5, 2\}$ for each task and report the best result. As shown in Tab. 6, CFG consistently improves the performance of LLaDA. We emphasize that, to ensure a fair comparison with ARMs, CFG is not used in the main results reported in the paper.

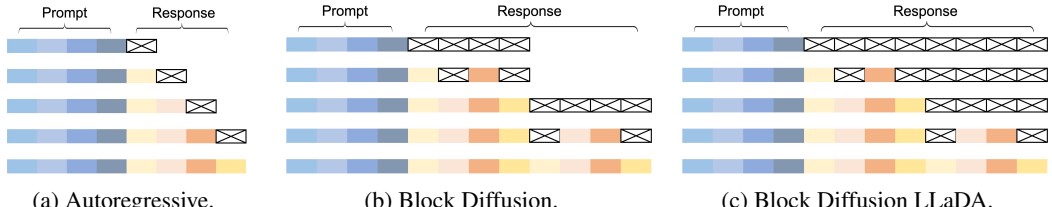

| | (a) Autoregressive. | (b) Block Diffusion. | (c) Block Diffusion LLaDA. |

Figure 4: **Flexible Sampling Strategies Supported by LLaDA.** Colored squares depict non-masked tokens, while squares marked with × denote masked tokens. In this illustration, the block length for both block diffusion and block diffusion LLaDA sampling is 4.

Table 7: **Ablation on Sampling Strategies for LLaDA 8B Base.** $L'$ is the block length. Pure diffusion sampling achieves the best overall performance.

| | | BBH | GSM8K | Math | HumanEval | MBPP |
|---|---|---|---|---|---|---|
| Autoregressive | | 38.1 | 63.1 | 23.6 | 18.3 | 33.4 |
| Block Difusion | $L' = 2$ | 37.3 | 62.6 | 25.2 | 14.6 | 33.6 |
| | $L' = 4$ | 40.0 | 65.7 | 26.6 | 15.9 | 36.0 |
| | $L' = 8$ | 42.0 | 68.2 | 27.7 | 19.5 | 39.2 |
| | $L' = 32$ | 45.7 | 68.6 | 29.7 | 29.9 | 37.4 |
| Block Diffusion LLaDA | $L' = 2$ | 48.0 | 70.0 | 30.8 | 26.2 | **40.0** |
| | $L' = 4$ | 48.5 | **70.3** | 31.3 | 27.4 | 38.8 |
| | $L' = 8$ | 48.6 | 70.2 | 30.9 | 31.1 | 39.0 |
| | $L' = 32$ | 48.3 | **70.3** | 31.2 | 32.3 | **40.0** |
| Pure Diffusion | | **49.7** | **70.3** | **31.4** | **35.4** | **40.0** |

Table 8: **Ablation on Sampling Strategies for LLaDA 8B Instruct.** The block length is set to 32 for efficiency. Pure diffusion sampling achieves the best overall performance.

| | GSM8K | Math | HumanEval | MBPP | GPQA | MMLU-Pro | ARC-C |
|---|---|---|---|---|---|---|---|
| Autoregressive | 0 | 9.5 | 0 | 0 | 0 | 0 | 84.4 |
| Block Diffusion | 24.6 | 23.5 | 17.1 | 21.2 | 29.3 | 32.5 | 88.1 |
| Block Difusion LLaDA | **77.5** | **42.2** | 46.3 | 34.2 | 31.3 | 34.8 | 85.4 |
| Pure Diffusion | 69.4 | 31.9 | **49.4** | **41.0** | **33.3** | **37.0** | **88.5** |

## B.4 Details and Ablation on Sampling Strategies

In this section, we first introduce the different sampling strategies supported by LLaDA. We then present ablation studies to evaluate the performance of these sampling strategies.

**Flexible Sampling Strategies.** In Sec. 2.4, Fig. 2 (c) illustrates the reverse generation process of LLaDA. As shown in Fig. 4, in addition to the reverse generation process, LLaDA also supports autoregressive and block diffusion [31] sampling directly after the pre-training or SFT stages, without requiring any further modifications or retraining. Block diffusion sampling applies the origin reverse process within each block and the autoregressive sampling across blocks. In the original block diffusion process, the sequence length varies dynamically. As shown in Fig. 4 (c), LLaDA can also adopt a fixed-length block diffusion strategy, which we refer to as block diffusion LLaDA, also known as semi-autoregressive remasking.

**Experimental Setup.** We evaluate different sampling strategies using both LLaDA 8B Base and LLaDA 8B Instruct for comprehensive analysis. For LLaDA 8B Base, we use the five benchmarks in Tab. 1 that are evaluated based on sampling rather than likelihood estimation. For LLaDA 8B Instruct, we use the seven metrics in Tab. 2, excluding MMLU and HellaSwag, since these two tasks only require the model to generate a single token (i.e., A, B, C, or D). In all settings, one token is generated

Table 9: **Analysis on Random and Low-confidence Remasking Strategies.** The low-confidence remasking consistently outperforms the random remasking.

| Length | BBH | GSM8K | Math | HumanEval | MBPP |
|---|---|---|---|---|---|
| Random Remasking | 32.1 | 21.3 | 9.2 | 11.6 | 21.0 |
| Low-confidence Remasking | **45.0** | **70.0** | **30.3** | **32.9** | **40.2** |

Table 10: **Ablation on Generation Length.** The results are not sensitive to the length hyperparameter.

| Length | BBH | GSM8K | Math | HumanEval | MBPP |
|---|---|---|---|---|---|
| 256 | 45.0 | 70.0 | 30.3 | 32.9 | **40.2** |
| 512 | **50.4** | **70.8** | 30.9 | 32.9 | 39.2 |
| 1024 | 49.7 | 70.3 | **31.4** | **35.4** | 40.0 |

per sampling step. For autoregressive and block diffusion sampling, generation terminates when the |EOS| token is produced. For block diffusion LLaDA (i.e., semi-autoregressive remasking) and pure diffusion sampling, the generation length is fixed at 1024 for LLaDA 8B Base, while for LLaDA 8B Instruct, it is tuned from {64, 256, 512} to balance efficiency and performance. Low-confidence remasking is applied to intra-block diffusion sampling in both block diffusion and block diffusion LLaDA, as well as to pure diffusion sampling. We also test different block lengths for LLaDA 8B Base. For LLaDA 8B Instruct, we only evaluate block length 32 for efficiency, as it yields the best results on LLaDA 8B Base.

Additionally, for LLaDA 8B Instruct, due to heavy padding of |EOS| tokens in the SFT data (as detailed in Sec. B.1), we observe that under pure diffusion sampling, the proportion of |EOS| tokens in model outputs becomes very high. This leads to extremely short generations and degrades model performance. To mitigate this issue, for HumanEval, MBPP, GSM8K, Math, and GPQA, we set the confidence score of the |EOS| token to zero during pure diffusion sampling. This adjustment helps maintain an appropriate ratio of |EOS| tokens during generation.

Finally, we conduct ablation studies to analyze the effects of random and low-confidence remasking strategies using the pure diffusion sampling. For efficiency, we use LLaDA 8B Base with generation length and sampling steps set to 256 in this analysis.

**Results.** As shown in Tab. 7, for block diffusion sampling, overall performance improves as the block length increases. Moreover, both Tab. 7 and Tab. 8 show that block diffusion sampling consistently outperforms autoregressive sampling, and block diffusion LLaDA sampling further improves upon standard block diffusion sampling. Finally, pure diffusion sampling achieves the best overall performance.

In addition, Tab. 9 shows that the low-confidence remasking strategy consistently outperforms the random remasking strategy. We hypothesize that low-confidence remasking functions similarly to the annealed sampling method used by default in ARMs, improving accuracy by reducing the diversity of generated sentences.

We discover that autoregressive sampling leads to very poor performance for LLaDA 8B Instruct. This is because each SFT data is a complete sentence, so given a sequence length, LLaDA 8B Instruct tends to generate a full sentence within that length. In contrast, LLaDA 8B Base does not suffer from this issue, as the pre-training data consists of truncated documents (as detailed in Appendix B.1) and the model is trained with random sequence lengths (as detailed in Sec. 2.2). As a result, when given a short sequence length, LLaDA 8B Base tends to generate only part of a sentence, which can then be used as a prefix to continue generation.

Setting the block length to 8 in Tab. 8 further improves the GSM8K score from 77.5 to 78.6.

## B.5 Ablation on Generated Length

In this section, we conduct ablation studies on the generated length.

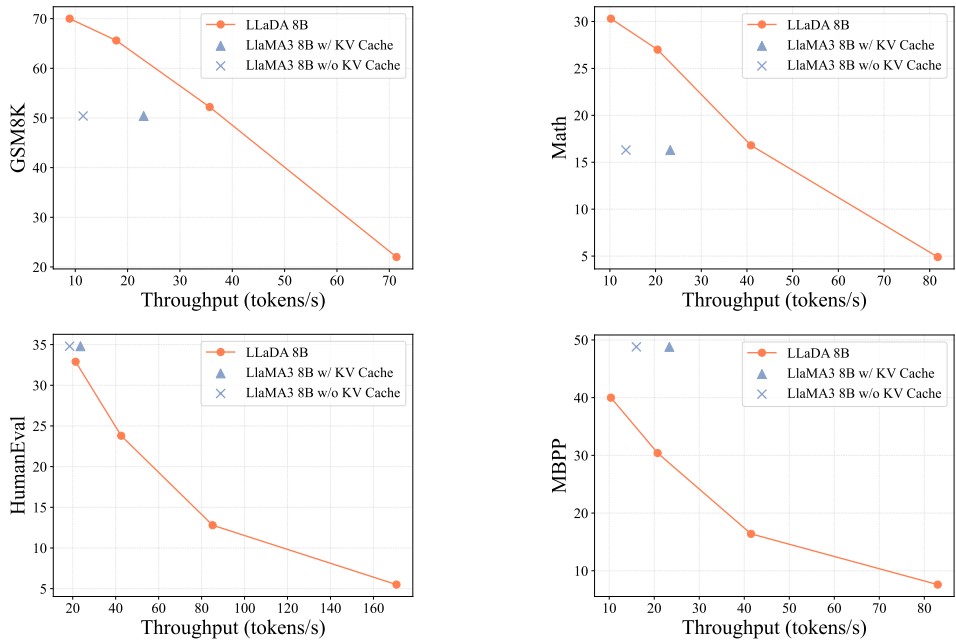

Figure 5: **Analysis of Sampling Efficiency.** The generation length for LLaDA is set to 256, with sampling steps set to 32, 64, 128, and 256 across the figures. This corresponds to decoding 8, 4, 2, and 1 token(s) per forward pass, respectively. LLaDA enables a flexible trade-off between generation quality and sampling speed.

To ensure fairness, for each setting, we set the number of sampling steps equal to the generated length, ensuring that in each sampling step, just one tokens are transferred from the mask to the text. We conduct experiments using LLaDA 8B Base.

As reported in Tab. 10, the results are not sensitive to the length hyperparameter.

## B.6 Standard Benchmarks and Evaluation Details

In this section, we introduce the benchmarks we used and present the details of our evaluation process.

Following standard LLM [25, 26] evaluation practices, we assess LLaDA across four dimensions:

**General ability:** MMLU [110], BBH [111], ARC-C [112], Hellaswag [113], TruthfulQA [114], WinoGrande [115] and PIQA [116].

**Math and science ability:** GSM8K [117], Math [118] and GPQA [119].

**Code generation:** HumanEval [120], HumanEval-FIM [121] and MBPP [122].

**Chinese understanding:** CMMLU [123] and C-Eval [124].

For all the aforementioned benchmarks, we follow the widely adopted evaluation process [125] used in LLM assessments, primarily employing conditional likelihood estimation and conditional generation. Specifically, for certain benchmarks, a prompt and multiple candidate answers are provided, and the model is required to compute each candidate's conditional likelihood. The candidate with the highest likelihood is then selected as the model's final answer, and accuracy is used as the evaluation metric. For the remaining benchmarks, the model generates responses based on the given prompt, and performance is evaluated using metrics such as exact match and other relevant criteria.

For the base model, we use conditional likelihood estimation for MMLU, CMMLU, C-Eval, ARC-C, Hellaswag, TruthfulQA, WinoGrande, PIQA, and GPQA, while the remaining benchmarks are evaluated using conditional generation. For the instruct model, we evaluate all benchmarks using conditional generation.

Table 11: **Analysis of Memory Consumption.** Memory is measured in GB. Without any inference optimization techniques (e.g., KV Cache), LLaDA has memory usage comparable to LLaMA3, and slightly higher than LLaMA3 when the latter uses KV Cache.

| Input Length | Output Length | LLaDA 8B | LLaMA3 8B w/o KV-Cache | LLaMA3 8B w/ KV-Cache |
|---|---|---|---|---|
| | 512 | 17.03 | 16.70 | 16.32 |
| 512 | 1024 | 17.53 | 17.49 | 16.43 |
| | 2048 | 18.52 | 20.00 | 16.73 |
| | 512 | 17.53 | 17.16 | 16.36 |
| 1024 | 1024 | 18.01 | 18.26 | 16.41 |
| | 2048 | 19.02 | 21.39 | 16.74 |

For the base model, we use the widely adopted open-source evaluation framework lm-evaluation-harness [125], except for the HumanEval-FIM metric, which is not supported by the framework. For HumanEval-FIM on the base model and all evaluation metrics on the instruct model, we use an internal evaluation library. We choose the internal library as lm-evaluation-harness shows greater deviation from the LLaMA3 results reported by Yang et al. [25], relative to our internal evaluation.

For benchmarks evaluated via conditional likelihood estimation, we use Monte Carlo estimation to approximate Eq. (6) for LLaDA. Since MMLU, CMMLU, and C-EVAL only require the likelihood of a single token, a single Monte Carlo estimate is sufficient for these benchmarks. For all other benchmarks, we find that 128 Monte Carlo samples are adequate to produce stable results.

For benchmarks evaluated using conditional generation, we apply pure diffusion sampling with a low-confidence remasking strategy to both LLaDA Base and LLaDA Instruct. For LLaDA Base, we set both the generation length and the number of sampling steps to 1024. For LLaDA Instruct, the number of sampling steps is set equal to the answer length, which is configured as follows: 3 for MMLU and HellaSwag, 64 for GPQA, 256 for MBPP and MMLU-Pro, and 512 for HumanEval, GSM8K, Math, and ARC-C. As detailed in Appendix B.4, for HumanEval, MBPP, GSM8K, Math, and GPQA, we set the confidence of the |EOS| token to zero during sampling for LLaDA Instruct.

## B.7 Analysis of Sampling Efficiency

In this section, we first analyze the sampling efficiency of LLaDA, including both sampling speed and memory consumption. We then discuss potential optimizations to further improve its efficiency.

We use four representative open-ended generation benchmarks for sampling speed analysis: GSM8K, Math, HumanEval, and MBPP. We use the widely adopted throughput metric to measure generation speed, defined as the number of tokens generated per second. We compare LLaDA 8B Base and LLaMA3 8B Base, both using bfloat16 precision. All experiments in this section were conducted on a single A100-80GB GPU with a batch size of 1. For LLaDA, the output length is fixed to 256 tokens across all four benchmarks.

Fig. 5 shows that LLaDA enables a flexible trade-off between generation quality and speed by adjusting the number of sampling steps. Specifically, on the GSM8K and Math datasets, LLaDA 8B Base achieves comparable performance to LLaMA3 8B Base while delivering 1.5 and 1.8 times higher throughput, even though LLaMA3 uses KV Cache and LLaDA operates without any inference optimization techniques. For the HumanEval benchmark, LLaDA 8B Base performs comparably to LLaMA3 8B Base when the throughput is matched. On the MBPP benchmark, LLaDA 8B Base lags behind LLaMA3 8B Base.

For LLaMA3, the acceleration benefit provided by KV caching is notably weaker on the HumanEval dataset, which can be attributed to its relatively short prompt lengths. Specifically, the average prompt lengths for GSM8K, Math, MBPP, and HumanEval are 894, 680, 628, and 132 tokens, respectively.

Tab. 11 compares of memory consumption between LLaDA 8B Base and LLaMA3 8B Base. To avoid variations in generation length caused by differences in training data, we fix both the input and output token lengths during the memory analysis. For LLaDA, memory usage remains constant regardless of the number of sampling steps. Its memory consumption is comparable to LLaMA3 8B Base without KV cache, but slightly higher than with KV cache.

Table 12: **Comparison on iGSM Dataset.**

|  | 4 steps | 5 steps | 6 steps |
| --- | --- | --- | --- |
| LLaMA3 8B Base | 38.0 | 35.0 | 34.0 |
| LLaDA 8B Base | **64.0** | **41.0** | **44.0** |

We emphasize that the goal of this study is not to propose a model that is faster than ARMs. Instead, we aim to show the promise of diffusion models for language modeling at scale and challenge the common assumption that core LLM capabilities such as scalability, in-context learning, and instruction-following are inherently depend on ARMs. A substantial body of research [31, 79–87] has focused on improving the generation efficiency of MDMs through algorithmic or architectural innovations. We leave similar efficiency-oriented exploration for LLaDA to future work.

## B.8 Evaluation on iGSM Dataset

To further assess the mathematical capabilities of LLaDA, we test its performance on iGSM [34], an infinite, synthetic GSM8K-like dataset. iGSM is generated via specific rules, with parameters that control the difficulty of problems (i.e., the number of solution steps). For evaluation consistency, we append "#### $answer" to the final solution, adhering to the GSM8K format. Below is an example with solution steps set to 4:

---

(**Question**) The number of each North Star Elementary's Cultural Studies Classroom equals 1. The number of each Westridge Elementary's Dance Studio equals 3 times as much as the sum of each North Star Elementary's Classroom and each North Star Elementary's Cultural Studies Classroom. How many Dance Studio does Westridge Elementary have?
(**Solution**) Define North Star Elementary's Cultural Studies Classroom as x; so x = 1.
Define North Star Elementary's Classroom as m; so m = x = 1.
Define Westridge Elementary's Dance Studio as n; w = m + x = 1 + 1 = 2;
so n = 3 * w = 3 * 2 = 1 #### 1

---

Since there are slight differences between GSM8K and iGSM (e.g., the use of a mod 5 algorithmic system), we follow [34] and provide a system prompt along with four-shot question-answer pairs for each problem.

---

(**Prompt**) You're an expert at solving elementary math problems involving addition, subtraction, and multiplication. You solve all the problems in a uniform format. All calculations are done modulo 5. For example, 4 + 4 equals 3, 2 + 4 equals 1, 3 + 3 + 3 equals 4, 3 * 3 equals 4, and 2 * 2 equals 4. When providing your solution, please end with '#### x.' where x is your final answer, an integer between 0 and 4. You must solve all the problems using the same solution format. Our scenarios involve up to four categories of objects: schools, classrooms, backpacks and stationeries. Each school may contain classrooms, each classroom may contain backpacks, and each backpack may contain stationeries. We can specify quantities, such as ẗhe number of dance studios at each Lakeshore High.Ässume that every entity with the same name has an identical configuration; for example, each Lakeshore High contains the same number of dance studios. Another guiding principle is that what is not mentioned does not exist: when we refer to classrooms at Lakeshore High, we are only discussing the classrooms explicitly mentioned in our scenario. Furthermore, if Lakeshore High is not even mentioned, any classroom within it is automatically considered to be non-existent (i.e. 0).

---

For solution steps ranging from 4 to 6, we generate 100 questions for each case and report the corresponding accuracy in Table 12. As shown in the table, LLaDA 8B Base demonstrates significant and consistent advantages over LLaMA3 8B Base on unseen mathematical problems, aligning with the results in Table 1.

Table 13: **Visualization of the Block Diffusion LLaDA Sampling Process.** In the response of LLaDA, darker colors indicate tokens predicted in the later stages of sampling, while lighter colors correspond to earlier predictions.

| User | What are the benefits of regular exercise for physical and mental health? |
| --- | --- |
| LLaDA | |

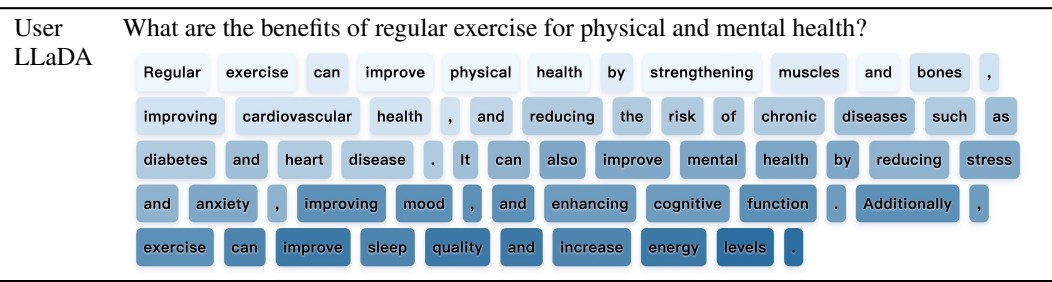

## B.9 Poem Completion Tasks

In this section, we present examples from our poem completion dataset as follows.

Example 1:
Prompt: 窈窕淑女的下一句是什么？直接输出句子即可。
Answer: 君子好逑。

Example 2:
Prompt: 不拘一格降人才的上一句是什么？直接输出句子即可。
Answer: 我劝天公重抖擞。

## B.10 More Case Studies

In this section, we present additional case studies of LLaDA 8B Instruct. First, Tab. 13 shows the sampling process of the block diffusion LLaDA sampling, while Tab. 14 depicts the sampling process for multi-turn dialogues with random remasking. Additionally, Tab. 15 and Tab. 16 provide further examples of single-turn and multi-turn dialogues. Finally, Tab. 17 presents examples of poem reversal completions where the LLaDA 8B Instruct model succeeds, in contrast to the failure of GPT-4o.

# C Impact Statement

Our work shows the promise of diffusion models for language modeling at scale and challenges the common assumption that core LLM capabilities such as scalability, in-context learning, and instruction-following are inherently dependent on ARMs. Our findings open new avenues for exploring alternative probabilistic paradigms in natural language processing, with potential applications in conversational AI, code generation, and complex reasoning tasks.

However, diffusion models, like traditional LLMs, raise similar societal concerns. These include the environmental impact of large-scale training, the potential misuse for generating harmful content, and the amplification of biases present in training data. Addressing these challenges is critical to ensuring the responsible development and deployment of diffusion language models.

Table 14: **Visualization of the Multi-turn Dialogue.** We employ random remasking strategy. In the response of LLaDA, darker colors indicate tokens predicted in the later stages of sampling, while lighter colors correspond to earlier predictions.

| User | I want to have a pet, but I don't know whether to get a cat or a dog. Do you have any suggestions? |
| --- | --- |
| LLaDA | 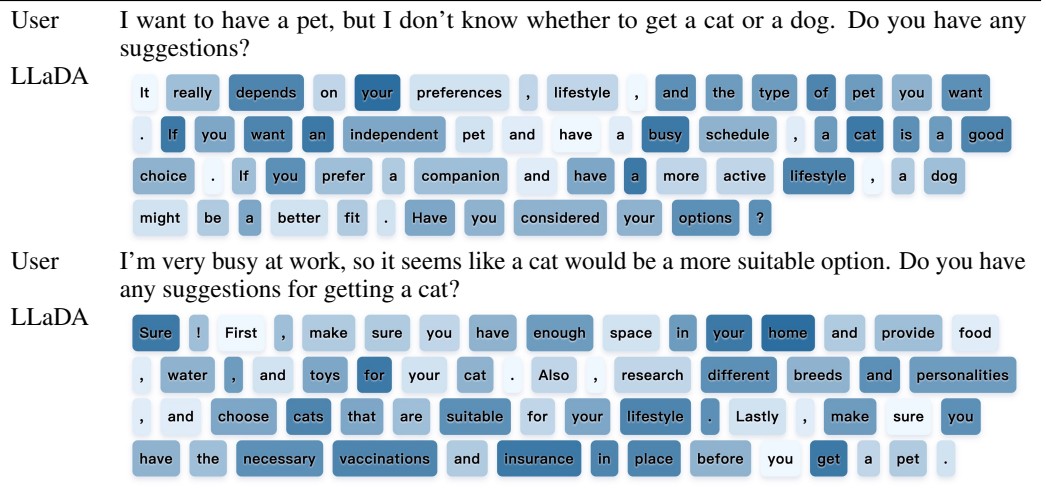 |
| User | I'm very busy at work, so it seems like a cat would be a more suitable option. Do you have any suggestions for getting a cat? |
| LLaDA | |

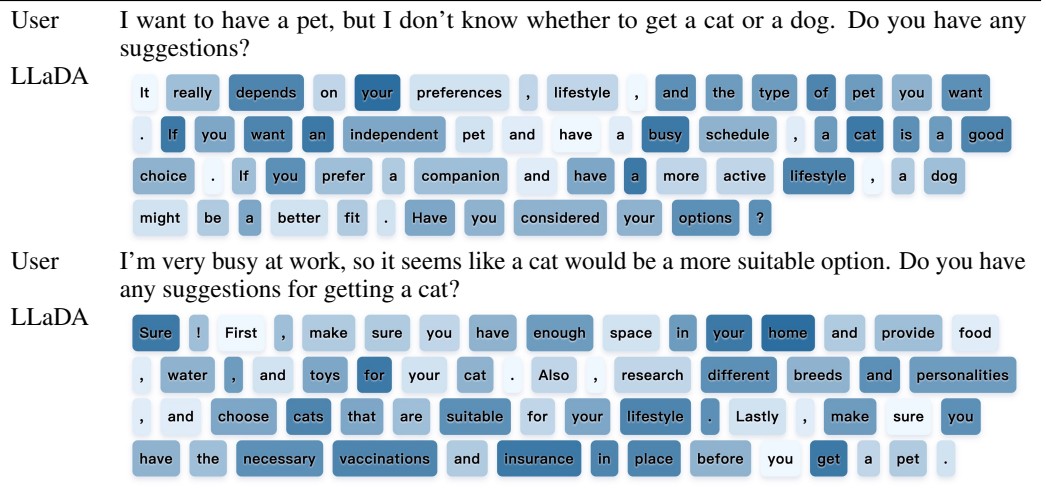

Table 15: **Single-turn Dialogue Cases of LLaDA 8B Instruct.**

| | |
|---|---|
| *Daily life* | |
| User | What's a piece of advice you've received that you've found helpful? |
| LLaDA | One helpful piece of advice I've received is to always be open to learning and growth. I've found that being willing to explore new ideas and experiences has helped me become more resilient and adaptable. By embracing a growth mindset, I've been able to identify opportunities for improvement and development, both personally and professionally. |
| *Daily life* | |
| User | Please recommend me three famous movies |
| LLaDA | Certainly! Here are three famous movies that have achieved widespread popularity and critical acclaim: |
| | 1. **The Shawshank Redemption** (1994) - Inspired by a novella and directed by Frank Darabont, this film tells the story of Andy Dufresne, a man convicted of murdering his wife, dealing with his harsh life in Shawshank Prison and forming an unlikely friendship with a fellow inmate. |
| | 2. **The Godfather** (1972) - Directed by Francis Ford Coppola, this film is based on Mario Puzo's novel of the same name. It follows the story of the Corleone family, particularly Michael Corleone, who rises to power and takes over the family's criminal empire. |
| | 3. **The Empire Strikes Back** (1980) - Directed by George Lucas, this film continues the story of a young Skywalker in the Star Wars saga. It follows Luke Skywalker as he helps the Rebel Alliance fight against the Galactic Empire and destroy the powerful weapon, the Death Star. |
| | All of these films are celebrated for their cinematic achievements, powerful storytelling, and memorable characters. |
| *Translation* | |
| User | Please help me translate into Chinese: 'What is now proved was once only imagined' |
| LLaDA | 现在被证明的曾经只是想象 |
| *Translation* | |
| User | Please help me translate into German: 'What is now proved was once only imagined' |
| LLaDA | Wer heute bestätigt ist, war einst nur immaginiert. |
| *Math* | |
| User | Emma decides to run 4 sprints 5 times a week. She runs 75 meters each sprint. How many total meters does she run a week? |
| LLaDA | Emma runs 4 sprints each time, so she runs a total of 4 x 5 = 20 sprints a week. Each sprint is 75 meters long, so Emma runs a total of 20 x 75 = 1500 meters a week. The final result is 1500 |
| *Code* | |
| User | I am writing python code, I have an array, please help me sort it from largest to smallest |
| LLaDA | Certainly! You can sort an array from the largest to smallest in Python using the 'sort()' method or the 'sorted()' function. Here's an example: |
| | ### Using 'sort()' method: |
| | my_array = [3, 1, 4, 1, 5, 2] |
| | my_array.sort(reverse=True) |
| | print(my_array) |

Table 16: **Multi-turn Dialogue Cases of LLaDA 8B Instruct.**

| | |
|---|---|
| *Daily life* | |
| User | How do I prepare for a job interview? |
| LLaDA | To prepare for a job interview, you should research the company and the job, practice common interview questions, and dress appropriately. |
| User | What are some common interview questions? |
| LLaDA | Some common interview questions include "Tell me about yourself" and "Why do you want to work for us"? |
| *Daily life* | |
| User | How do I start learning programming? |
| LLaDA | To learn programming, you can start by choosing a programming language that you are interested in. Some popular languages include Python, Java, and C++. |
| User | Which programming language should I learn first? |
| LLaDA | It depends on your interests and goals. Python is a great language for beginners for its simplicity and versatility. |
| User | How long does it take to learn Python? |
| LLaDA | It can take anywhere from a few weeks to several months to learn Python. |
| User | What are some good Python projects for beginners? |
| LLaDA | Some good Python projects for beginners include building a simple calculator, a game, and a web application. |
| *Math* | |
| User | Natalia sold clips to 48 of her friends in April, and then she sold half as many clips in May. How many clips did Natalia sell altogether in April and May? |
| LLaDA | Natalia sold clips to 48 friends in April. She sold half as many clips in May, so she sold 48/2 = 24 clips in May. Altogether, Natalia sold 48 + 24 = 72 clips in April and May. The final result is 72. |
| User | Natalia sold three times as many items in June as she did in May. How many clips did Natalia sell altogether in these three months? |
| LLaDA | Natalia sold clips to 48 friends in April. She sold half as many clips in May, so she sold 48/2 = 24 clips in May. Natalia sold three times as many items in June as she did in May, so she sold 24 * 3 = 72 clips in June. Altogether, Natalia sold 48 + 24 + 72 = 144 clips in these three months. The final result is 144. |

Table 17: **Poem Reversal Completion Cases where LLaDA 8B Instruct Succeeds but GPT-4o Fails.**

| | |
|---|---|
| *Question1* | |
| User | '但闻人语响'的上一句是什么? |
| GPT-4o | "但闻人语响"的上一句是"无可奈何花落去",这两句出自唐代词人李清照的（武陵春,春晚:无可奈何花落去,但闻人语响。这首词表达了作者在春天时节的孤独与感伤,借花落和人语声的描写,反映出词人内心的无亲与失落。 |
| LLaDA | "但闻人语响"的上一句是"空山不见人" |
| *Question2* | |
| User | '夜静春山空'的上一句是什么? |
| GPT-4o | "夜静春山空"的上一句是"空山新雨后".这两句出自唐代诗人王维的（山居秋暝）:空山新雨后,夜静春山空。这首诗描绘了春天山中雨后的寂静与空灵,表现出自然的宁静与和谐美。 |
| LLaDA | "夜静春山空"的上一句是"人闲桂花落" |

Table 18: **Detailed results of LLaDA in Fig. 3.** "-" indicates missing values, which do not affect the observations regarding the scalability of LLaDA. These missing values are due to hardware failures.

| Model | Training Tokens | FLOPs | MMLU | CMMLU | ARC-C | PIQA | GSM8K | HumanEval |
|---|---|---|---|---|---|---|---|---|
| LLaDA 1B | 37.75B | 2.20e20 | 25.52 | 25.95 | 25.17 | 59.41 | 1.82 | 0.00 |
| LLaDA 1B | 88.08B | 5.13e20 | 27.11 | 26.52 | 26.96 | 61.86 | 3.03 | 1.83 |
| LLaDA 1B | 138.41B | 8.06e20 | 29.32 | 27.13 | 30.20 | 63.38 | 2.35 | 0.00 |
| LLaDA 1B | 239.08B | 1.39e21 | 31.48 | 30.77 | 27.99 | 63.11 | 3.26 | 1.22 |
| LLaDA 1B | 352.32B | 2.05e21 | 35.86 | 34.35 | 31.31 | 65.34 | 3.64 | 3.05 |
| LLaDA 1B | 461.37B | 2.69e21 | 31.86 | 30.98 | 30.12 | 65.51 | 2.35 | 0.61 |
| LLaDA 8B | 62.91B | 2.63e21 | 32.22 | 28.5 | 30.20 | 63.82 | 3.87 | 2.44 |
| LLaDA 8B | 125.83B | 5.27e21 | 33.39 | 33.9 | 34.64 | 66.54 | 8.72 | 3.66 |
| LLaDA 8B | 251.66B | 1.05e22 | 42.84 | 40.59 | 40.10 | 69.04 | 15.31 | 3.66 |
| LLaDA 8B | 377.49B | 1.58e22 | 45.11 | 43.99 | 39.25 | 68.61 | 25.40 | 9.76 |
| LLaDA 8B | 503.32B | 2.11e22 | 43.57 | 41.38 | 42.06 | 70.24 | 27.52 | 9.76 |
| LLaDA 8B | 629.14B | 2.63e22 | 48.80 | 47.13 | 42.24 | 72.09 | 30.10 | 12.80 |
| LLaDA 8B | 679.48B | 2.85e22 | 49.61 | 48.19 | 41.30 | 70.84 | 26.31 | 8.54 |
| LLaDA 8B | 792.72B | 3.31e22 | 50.88 | 49.01 | 42.58 | 70.51 | 31.99 | 6.10 |
| LLaDA 8B | 981.47B | 4.11e22 | 49.47 | 48.10 | 40.27 | 71.38 | - | 6.10 |
| LLaDA 8B | 1107.30B | 4.64e22 | 51.13 | 47.57 | 41.13 | 69.26 | 36.69 | 10.37 |
| LLaDA 8B | 1233.13B | 5.16e22 | 50.52 | 49.72 | 45.05 | 71.49 | 38.97 | 9.76 |
| LLaDA 8B | 1358.95B | 5.69e22 | 54.61 | 53.97 | 49.40 | 74.05 | 48.14 | 17.68 |
| LLaDA 8B | 1547.70B | 6.48e22 | 57.38 | 56.04 | 49.49 | 74.59 | 53.30 | 20.73 |
| LLaDA 8B | 1975.52B | 8.27e22 | 58.52 | 57.87 | 50.68 | 75.35 | - | 19.51 |

Table 19: **Detailed results of the autoregressive baelines in Fig. 3.**

| Model | Training Tokens | FLOPs | MMLU | CMMLU | ARC-C | PIQA | GSM8K | HumanEval |
|---|---|---|---|---|---|---|---|---|
| ARM 1B | 37.75B | 2.20e20 | 25.47 | 25.38 | 30.20 | 67.36 | 2.20 | 4.88 |
| ARM 1B | 88.08B | 5.13e20 | 24.67 | 25.23 | 33.96 | 70.02 | 7.51 | 10.37 |
| ARM 1B | 138.41B | 8.06e20 | 29.25 | 27.48 | 33.45 | 70.29 | 8.34 | 9.76 |
| ARM 7B | 17.30B | 6.02e20 | 26.92 | 25.18 | 21.02 | 57.18 | 1.29 | 1.22 |
| ARM 7B | 34.60B | 1.20e21 | 25.83 | 25.38 | 24.07 | 62.84 | 1.59 | 2.44 |
| ARM 7B | 86.50B | 3.01e21 | 24.41 | 24.90 | 25.42 | 71.11 | 2.88 | 7.93 |
| ARM 7B | 173.02B | 6.02e21 | 26.20 | 24.78 | 26.10 | 74.27 | 6.67 | 9.15 |
| ARM 7B | 207.62B | 7.23e21 | 30.36 | 28.86 | 31.86 | 74.48 | 8.57 | 12.80 |
| ARM 7B | 224.92B | 7.83e21 | 29.49 | 32.26 | 31.19 | 74.37 | 8.95 | 8.54 |
| ARM 7B | 242.22B | 8.43e21 | 33.62 | 31.38 | 34.92 | 75.41 | 10.84 | 9.15 |
| ARM 7B | 259.52B | 9.03e21 | 34.11 | 34.20 | 32.88 | 75.19 | 9.33 | 10.98 |
| ARM 7B | 311.43B | 1.08e22 | 35.66 | 35.49 | 36.61 | 75.14 | 11.30 | 10.37 |
| ARM 7B | 363.33B | 1.26e22 | 34.54 | 37.67 | 34.58 | 76.55 | 12.28 | 14.02 |
| ARM 7B | 415.24B | 1.45e22 | 35.37 | 38.37 | 35.25 | 76.39 | 14.40 | 12.80 |
| ARM 7B | 449.84B | 1.57e22 | 39.51 | 39.24 | 34.92 | 76.82 | 14.94 | 14.63 |
| ARM 7B | 519.09B | 1.81e22 | 40.30 | 40.69 | 37.29 | 77.15 | 14.03 | 14.63 |
| ARM 7B | 778.57B | 2.71e22 | 43.33 | 43.50 | 38.31 | 77.53 | 17.59 | 14.63 |
| ARM 7B | 1038.09B | 3.61e22 | 45.06 | 46.12 | 41.69 | 77.86 | 20.02 | 15.85 |
| ARM 7B | 1384.12B | 4.82e22 | 47.63 | 48.18 | 47.80 | 76.93 | 22.82 | 15.24 |
| ARM 7B | 2076.18B | 7.23e22 | 47.68 | 50.85 | 44.07 | 77.37 | 24.79 | 14.63 |
| ARM 7B | 2214.59B | 7.71e22 | 49.26 | 52.08 | 53.56 | 77.69 | 27.37 | 17.07 |

