# OpenReview forum: "Large Language Diffusion Models"
_NeurIPS.cc/2025/Conference — NeurIPS 2025 oral_

### Official Review · Reviewer_APN5 · 2025-06-04

**Clarity:** 3
**Significance:** 3
**Originality:** 3
**Rating:** 5
**Confidence:** 5

**Summary:**

This paper introduces LLaDA, a diffusion language model trained from scratch. LLaDA shows comparable performances with auto-regressive models under the same experiment settings. After complete training process including Pre-training and SFT, LLaDA 8B is comparable with LLaMA3 8B. Moreover, LLaDA has advantages on some capabilities, such as the reversal curse, than auto-regressive models. LLaDA shows a promising direction for diffusion language models as general LLM.

**Questions:**

1. The key difference between language and image is in the variational length. Although the generation length is easy to decide for most of the benchmarks, it should still be fixed for real deployment. Is there any elegant solution for this?

2. It seems LLaDA requires more FLOPs than ARM baselines. However, in Figure 5, the throughput of LLaDA is much better than ARM. Is there any optimization for ARM? After all, ARM decoding is memory-bound while LLaDA seems to be computation-bound. It's also weird that w/ KV cache and w/o KV cache's throughput are quite similar.

3. To my understanding, there is a misalignment between training and inference on LLaDA. In training, all of the tokens are treated equally, while in inference, the prompt is given as a condition and always the clean token. Could you give any discussion for this?

**Ethical Concerns:**

["NO or VERY MINOR ethics concerns only"]

**Final Justification:**

The author solves most of my concerns. I believe it is a good work.

**Limitations:**

Limitations are discussed in the paper.

**Paper Formatting Concerns:**

No concerns.

**Quality:**

3

**Strengths And Weaknesses:**

Strengths:
1. LLaDA is the first strong pre-trained model comparable with well-known auto-regressive language models.
2. The model recipe is simple and elegant, without any ad-hoc configuration and cherry-pick evaluation.
3. Diffusion language model shows a capability advantage over auto-regressive models.

Weaknesses:
There is no obvious weakness in the paper. However, as a new proposed model design rather than a pre-training technical report, most of the experiment part should focus on the comparison with baseline models. However, only Section 3.1 makes a fair comparison, the rest of the pair is on the comparison with other open source models. Although the performance is also strong, the **fair comparsion** is better to present the effectiveness of the proposed method.

---

> ### Author Rebuttal · Authors · 2025-07-31
>
> # Response to Reviewer APN5
> We thank Reviewer APN5 for the recognition of our contributions and the thoughtful comments. Below is our point-by-point response.
>
>
> ## Q1: Comparison with Autoregressive Baselines
> We appreciate your recognition of the fair comparison presented in Sec. 3.1. Specifically, we evaluate LLaDA and autoregressive baselines across six diverse and representative tasks, including general language understanding, mathematics, code generation, and Chinese. The comparison spans model sizes from 1B to 8B parameters and pre-training data scales from 30B to 2T tokens, covering a total of 40 checkpoints. In total, **we conducted 240 experimental configurations, enabling a broad and thorough comparison between LLaDA and autoregressive baselines across different tasks, model sizes, and data scales.**.
>
>
> After establishing a fair and comprehensive comparison with autoregressive baselines (in Sec. 3.1), we follow a common practice in the field by further comparing LLaDA with well-known open-source models (in Sec. 3.2). In Sec. 3.3, to highlight LLaDA’s advantage in reversal reasoning, we compare it with GPT-4o, a state-of-the-art model that far surpasses our autoregressive baseline. We believe this represents a stronger benchmark, and better highlights LLaDA’s advantage in reversal reasoning. Finally, Sec. 3.4 presents case studies.
>
>
> ## Q2: Output Length
> Thanks for your question. In response to your inquiry about an elegant solution, we believe that jointly modeling the response and its length offers a promising and elegant approach. Specifically, let $p$ denote the prompt, $r$ the response, and $l$ the response length. We consider adding a lightweight head to the Transformer to predict the response length, and model $p_{\theta}(r, l|p)$ using maximum likelihood estimation as the objective function. We leave the implementation details to future work.
>
>
> ## Q3: Throughput
> Thanks for your question. Below, we would first like to clarify LLaDA’s throughput advantage, and then explain the efficiency differences caused by KV cache for autoregressive models.
>
>
> **The parallel decoding capability is the key reason behind LLaDA’s advantage in sampling efficiency on certain benchmarks.** As detailed in Fig. 5 (Page 25), on the GSM8K and Math benchmarks, when **each forward pass generates 4 tokens** (i.e., the number of sampling steps is set to one-fourth of the generation length), LLaDA achieves slightly better accuracy than LLaMA3 with KV Cache, while delivering 1.5× and 1.8× higher throughput, respectively.
>
>
> The acceleration effect of KV cache for autoregressive models depends on both the sequence length and the hardware resources. As described in Lines 868 to 870, all efficiency analysis experiments were conducted on a single A100 GPU with 80GB of memory, using a batch size of 1. Both LLaMA3 and LLaDA operated with bfloat16 precision. For the HumanEval benchmark, where the average prompt length is relatively short at 132 tokens, enabling KV cache in LLaMA3 leads to a modest speedup of 1.27 times. In contrast, for benchmarks with longer prompts such as GSM8K, Math, and MBPP, whose average prompt lengths are 894, 680, and 628 tokens respectively, KV cache yields more significant improvements, with speedups of 2.09, 1.72, and 1.46 times accordingly.
>
>
> ## Q4: Is there a train-inference misalignment?
> Thank you for your question. **The inference setup is consistent with the SFT stage and can be regarded as a special case of the pre-training phase.** During unsupervised pre-training, the model learns to model an unconditional distribution and therefore treats all tokens equally. During SFT, it models a conditional distribution, so the prompt is kept clean and the loss is computed only on the response. At inference time, the model samples from a conditional distribution, and therefore the prompt is kept clean as well.
>
> In comparison, this process is consistent with that of autoregressive models, which also treat all tokens equally during pre-training but **compute the loss only on the response during SFT**. During inference, they likewise sample from a conditional distribution, generating from the prompt as in the SFT stage, rather than generating unconditionally as in pre-training.
>
> **If you have any more questions, we are happy to discuss them and will do our best to address them!**

---

> > ### Comment · Reviewer_APN5 · 2025-08-03
> >
> > Thank you for the detailed and thoughtful responses.
> > 1. The mention of “240 experimental configurations” sounds particularly impressive! It would be great to see this experimental matrix released publicly — perhaps as a supplementary table or appendix — as it would really strengthen the comprehensiveness of the comparison and allow others to build on your work more easily.
> > 2. Modeling output length as a prediction is a very interesting direction. I’m looking forward to seeing how you develop this idea in future work.
> > 3. Regarding throughput, I fully agree with your explanation. In addition, I believe it would be valuable to explore throughput more systematically across a wider range of input and output lengths — especially in light of emerging needs around long-context processing and test-time scaling.
> >
> > I don’t have further questions. For a contribution that is somewhat pioneering in nature, it’s understandable that some implementation details are left for future work.

---

> ### Author Response · Authors · 2025-08-03
>
> Thank you very much for your reply. We truly appreciate your recognition of our rebuttal and your comment that our work is pioneering in nature.
>
> In the revised version, we will include the experimental matrix detailing the 240 experimental configurations.
> We fully agree with your view that long-context processing and test-time scaling are important future directions, and we will continue to explore them in our future work.
>
> We believe that your constructive feedback has significantly improved the clarity and overall quality of the paper.

---

### Official Review · Reviewer_mRMF · 2025-06-23

**Clarity:** 4
**Significance:** 3
**Originality:** 2
**Rating:** 5
**Confidence:** 2

**Summary:**

This paper develop language models based on diffusion models rather than autoregressive models. The paper examines this under scaling, demonstrating similar performance to equally sized autoregressive LMs, and improved performance on the reversal curse problem.

**Questions:**

- Outside of the reversal curse, what advantages are there to using diffusion models? Why might they have different capability profiles than their autoregressive counterparts?
- .Can the authors comment on the novelty of the paper with regards to other diffusion models such as Inception's Mercury?
- I think the paper would benefit from a more extensive related work section as some fundamental work in a similar vein is missing, e.g., https://arxiv.org/abs/2405.20519

**Ethical Concerns:**

["NO or VERY MINOR ethics concerns only"]

**Final Justification:**

This is a very strong paper. I still think it's a clear accept. The authors have engaged with comments meaningfully and offered to make alterations to improve the paper.

**Limitations:**

Mostly yes, however, it would be interesting for the authors to comments on limitations of diffusion language models (rather than just the limitations of their own experiments (which they did well)).

**Quality:**

3

**Strengths And Weaknesses:**

Strengths:
- The paper is well written, clear, and succinct. I found it particularly easy to follow what the authors are describing.
- There are clear advantages to diffusion models over autoregressive models (e.g., outperforming on tasks like the reversal curse).
- The comparison from autoregressive models to diffusion models was done well, with a fair comparison based on identical model size and training data. Additionally, the evaluation of LLaDA as a tool was also strong, with comparisons presented against reasonable alternatives (e.g., Llama) despite differences in training data or exact size.

Mixed:
- It's unclear how novel the approach is. The core of the method is the mask predictor and its role in the model's loss function. I have insufficient expertise in diffusion models to know whether this is a standard approach or truly novel. Nevertheless, the end result or application is seemingly novel. However, there are other diffusion language models now available. e.g., Mercury from https://www.inceptionlabs.ai/. It looks to me as those these have been contemporaneously developed -- yet they should probably be discussed in the related work section.

Weaknesses:
- When LLaDA underperforms against other models (e.g., Llama) it can be a significant amount (e.g., as seen in Table 2). On the other hand, when LLaDA outperforms other models it's often marginally. This doesn't seem to be discussed in detail. More broadly, the paper doesn't explore or speculate on why diffusion models may have different capability profiles than their autoregressive counterparts (outside of the reversal curse).

---

> ### Author Rebuttal · Authors · 2025-07-31
>
> # Response to Reviewer mRMF
> We thank Reviewer mRMF for the recognition of our contributions and the thoughtful comments. Below is our point-by-point response.
>
>
> ## Q1: Discussion about Inception's Mercury
> Thank you for your question. **The Mercury [1] technical report does not provide details on the model architecture, training loss, inference method, or other technical aspects.** Specifically, Section 2.1 of [1] briefly mentions that Mercury adopts a forward noising process and a reverse denoising process, but it does not describe the specific formulations of these processes or the loss function. In addition, with respect to the model architecture, [1] only states that Mercury uses a Transformer, without clarifying whether it is bidirectional or causal, or whether it follows the standard Transformer design or a modified Diffusion Transformer.
>
> In contrast, our submission provides detailed information on model training and inference, and we will release our model weights, evaluation code, and training code.
>
> Considering the novelty, we kindly note that new paradigms, conceptual advances, or novel discoveries also reflect forms of scientific innovation. Our findings first show the promise of diffusion models for language modeling at scale and challenge the common assumption that core LLM capabilities including scalability, in-context learning, and instruction following are inherently depend on autoregressive models. Other reviewers, such as Reviewer YafR, also recognized our contribution and commented that our work “marks a significant contribution to the community” and “highlights the potential of non-AR approaches and broadens the design space for future language model development.”
>
> **Following your suggestion, we will include a discussion of the concurrent work Mercury [1] in the related work of the revised version.**
>
> [1] Inception Labs. Mercury: Ultra-Fast Language Models Based on Diffusion. arXiv 2025.06. https://arxiv.org/pdf/2506.17298.
>
>
> ## Q2: Comparison with Autoregressive Models
> Thank you for your question. In the following, we first explain the reasons behind the performance differences between LLaDA and LLaMA3, and then analyze the unique characteristics of LLaDA.
>
> ### Performance differences with LLaMA3
> LLaDA underperforms LLaMA3 on certain benchmarks, which is primarily due to differences in training data scale and the use of reinforcement learning. Specifically, LLaMA3 was pretrained on 15T tokens, while LLaDA was trained on only 2.3T tokens. In addition, LLaMA3 was aligned using reinforcement learning, whereas LLaDA has not yet undergone any reinforcement learning alignment.
>
> That said, we note that LLaDA significantly outperforms LLaMA3 on several benchmarks, such as GSM8K and Math, as shown in Tab. 1 (on Page 7). Moreover, we emphasize that **when trained on the same dataset, LLaDA is overall comparable to autoregressive baselines**, which follow the LLaMA3 architecture (see Fig. 3 on Page 6).
>
>
> ### Unique characteristics of LLaDA
>
> When modeling the joint distribution of data, autoregressive models optimize only a single factorization order—typically from left to right. In contrast, **LLaDA is trained to consider multiple conditioning directions, which may provide greater flexibility and lead to better generalization.** The advantage of LLaDA on reversal reasoning is a concrete example of this property.
>
>
> We further speculate that bidirectional training might also offer benefits in other tasks such as mathematical reasoning and code generation. The experimental results presented in Lines 775–821, where autoregressive sampling underperforms diffusion sampling for LLaDA, offer valuable insight into this hypothesis. However, compared to reversal reasoning, these tasks are more complex, making it more difficult to design experiments that directly isolate the benefits of bidirectionality. We leave a more thorough investigation to future work.
>
>
>
> ## Q3: Advantages of Diffusion Language Models
> Thank you for your question. Building on the bidirectional modeling capability of LLaDA, we would like to point out several additional advantages, beyond its strong performance on reversal reasoning as demonstrated in our submission.
>
> 1. LLaDA has the potential to achieve faster sampling via parallel generation, as inspired by Fig. 5 on Page 25.
> 2. As our work has demonstrated the potential of diffusion models for text generation, and considering their proven success in visual domains, LLaDA may be particularly well-suited to support future multimodal integration.
> 3. Building on the hypothesis that bidirectional modeling may offer better generalization, diffusion language models could have an advantage over autoregressive models when data is scarce [2].
>
>
> [2] Prabhudesai et al. Diffusion Beats Autoregressive in Data-Constrained Settings. arXiv 2025.07.
>
>
>
> ## Q4: More Related Work
> Thanks for your question. Reference [3], mentioned in your comment, proposes a program synthesis approach using a syntax tree-guided diffusion model, jointly trained with a value model to guide the denoising process. It shows strong performance on inverse graphics tasks. Following your suggestion, we will include a discussion of this work in the related work of the revised version.
>
> [3] Kapur et al. Diffusion On Syntax Trees For Program Synthesis. ICLR 2025.
>
>
> ## Q5: Limitations of Diffusion Language Models
> Thanks for your comment that we did well in discussing the limitations of our experiments.
> Regarding the broader limitations of diffusion language models, one issue is that their architectures do not naturally support inference acceleration techniques such as KV cache. While there has been some recent work [4] exploring KV cache mechanisms for diffusion models, these approaches typically introduce small approximation errors. We will add this discussion in the revised version.
>
> [4] Wu et al. Fast-dLLM: Training-free Acceleration of Diffusion LLM by Enabling KV Cache and Parallel Decoding. arXiv 2025.05.
>
> **If you have any more questions, we are happy to discuss them and will do our best to address them!**

---

> > ### Comment · Reviewer_mRMF · 2025-08-05
> >
> > Thank you for the thorough response to my questions.
> >
> > If the additions that you claim will be added are indeed added then I believe the paper will be stronger, and I am happy to support acceptance.

---

> > > ### Author Response · Authors · 2025-08-06
> > >
> > > We sincerely appreciate your recognition of our contribution. Your insightful feedback has undoubtedly enhanced the quality of our work, and we are committed to incorporating the relevant discussions from the rebuttal into the final version of the paper.

---

### Official Review · Reviewer_vGqQ · 2025-07-03

**Clarity:** 3
**Significance:** 3
**Originality:** 2
**Rating:** 5
**Confidence:** 4

**Summary:**

This paper scales up the training for diffusion language model to more than 2T pretraining tokens and 8B model size, achieving superior performance than the autoregressive counterpart across a broad range of downstream benchmarks. The artifact, LLaDA, shows strong capabilities in scalability, in-context learning, and instruction-following, overcoming the limitations of autoregressive models on certain tasks such as the reversal curse.

**Questions:**

1. Could the author explain how is the output length controlled during evaluation? The paper says that <EOS> tokens were injected during pretraining and removed during sampling. However, I did not understand how the length is predicted at inference time--do you set a fixed length and force the model to fill the length for different samples, or do you ask the model to generate a maximum length and cut after <EOS>? How is the output length predicted for different evaluation samples?
2. Could you explain more about how you did conditional likelihood evaluation? What is a lower variance and upper bound in 2.4 and how they are used?
3. Do you have any insights on why diffusion LM outperforms autoregressive LMs with the same training FLOPs? Do you think your pretraining data quality is higher than the baseline models?
4. Do you plan to open-source your data and training code?

**Ethical Concerns:**

["NO or VERY MINOR ethics concerns only"]

**Final Justification:**

This paper present comprehensive experiments for scaling up the training for diffusion LMs, providing meaningful insights for practitioners. My questions have been addressed during rebuttal and I have raised my score accordingly.

**Limitations:**

yes.

**Quality:**

4

**Strengths And Weaknesses:**

Strenghts:
1. The performance is impressive--it is the first work showing that the diffusion language model could performance comparable and even superior than well-established autoregressive models.
2. Comprehensive evaluation is conducted to show the strong capabilities of the trained diffusion model on various tasks.

Weaknesses:
1. There is limited novelty on architecture design and training method. The main contribution is on scaling up the training set.
2. The paper writing could be further improved to clarify some details (see questions).

---

> ### Author Rebuttal · Authors · 2025-07-31
>
> # Response to Reviewer vGqQ
> We thank Reviewer vGqQ for the recognition of our contributions and the thoughtful comments. Below is our point-by-point response.
>
> ## Q1: Contribution
> Thank you for your question. We kindly note that, in addition to the architectural design and training methods you mentioned, new paradigms, conceptual advances, or novel discoveries also reflect forms of scientific innovation. We illustrate this point using two highly influential examples: Vision Transformer (ViT) [1] and GPT-3 [2].
>
> As early as ICLR 2020, [1] proposed the idea of applying Transformers to the image recognition domain. ViT, published at ICLR 2021, further demonstrated that large-scale pre-training can make vanilla Transformers competitive with, or even superior to, state-of-the-art CNNs. Similarly, GPT-3 builds on the same architecture and generative pre-training approach introduced in GPT-1 and GPT-2, and shows that scaling up language models significantly improves task-agnostic, few-shot performance. **Neither ViT nor GPT-3 introduced fundamentally new architectures or training algorithms, yet both have profoundly shaped the current landscape of artificial intelligence.**
>
>
> Our findings first show the promise of diffusion models for language modeling at scale and challenge the common assumption that core LLM capabilities including scalability, in-context learning, and instruction following are inherently depend on autoregressive models. We appreciate your comment, "it is the first work showing that the diffusion language model could perform comparably to, and even better than, well-established autoregressive models".
>
> We will include the above discussion in the revised version.
>
>
> [1] Dosovitskiy et al. An Image is Worth 16x16 Words: Transformers for Image Recognition at Scale. ICLR 2021.
>
> [2] Brown et al. Language Models are Few-Shot Learners. NeurIPS 2020.
>
> [3] Cordonnier et al. On the Relationship between Self-Attention and Convolutional Layers. ICLR 2020.
>
>
> ## Q2: Output Length
> Thank you for your question. In your comment, you mentioned that "The paper says that [object Object] tokens were injected during pretraining." We assume that the "[object Object]" text is a typographical error, and based on the context, it likely refers to the |EOS| token. If our interpretation is incorrect, please feel free to let us know.
>
> As detailed in Lines 150–152, the generation length is treated as a hyperparameter that specifies the length of the fully masked sentence at the beginning of the sampling process. Consequently, the length of the sentence generated by LLaDA corresponds to this hyperparameter. To support variable-length generation, we truncate the output at the |EOS| token.
>
> We analyze the impact of this hyperparameter, as shown in Tab. 10 (Page 24). For generation lengths of 256, 512, and 1024, the maximum performance differences observed on the BBH, GSM8K, Math, HumanEval, and MBPP benchmarks are 5.4, 0.8, 1.1, 2.5, and 1.0 points, respectively. Overall, the impact is limited and remains well-controlled.
>
> ## Q3: Conditional Likelihood Evaluation
> Thank you for your question. As detailed in Eq. (4) of our submission, the training loss of LLaDA has been proven to be an upper bound on the negative log-likelihood of the model distribution. Please refer to the Appendix A of [4] for the proof.
>
> For conditional likelihood estimation, we derive two upper bounds, namely Eq. (5) and Eq. (6). Since the expectations in both equations are intractable to compute analytically, we estimate them using Monte Carlo sampling. While both estimators yield the same expected value, the one based on Eq. (6) exhibits lower variance, as it avoids the additional randomness introduced by sampling $r_t$. Please refer to Lines 694–700 for a more detailed discussion.
>
> [4] Sahoo et al. Simple and Effective Masked Diffusion Language Models. NeurIPS 2024.
>
>
> ## Q4: Insight on Performance Advantage
> Thanks for your question. **Both LLaDA and our autoregressive baselines (in Fig. 3 on Page 6) are trained on the same dataset**, as detailed in Lines 176–179. As for LLaDA’s performance advantages, some possible explanations are as follows:
>
> (1) When modeling the joint distribution of data, autoregressive models optimize only a single factorization order—typically from left to right. In contrast, LLaDA is trained to consider multiple conditioning directions, which may provide greater flexibility and lead to better generalization. The advantage of LLaDA on reversal reasoning is a concrete example of this property.
>
> (2) We hypothesize that removing the inductive bias of left-to-right generation in text and allowing the model to learn the generation order on its own may benefit model training. A similar observation was made in the Vision Transformer (ViT) paper [1], which showed that removing certain inductive biases from the data (e.g., translation equivariance in images) can enable Transformers to learn more effectively.
>
>
> ## Q5: Open-Source
>
> Thank you for your question. We will release our model weights, evaluation code, and training code. Details of our data collection process are provided in Lines 723–744. However, similar to several influential works [5, 6, 7], we choose not to release the training data to mitigate risks associated with the misuse of sensitive content or potential data biases.
>
> [5] Saharia et al. Photorealistic Text-to-Image Diffusion Models with Deep Language Understanding. NeurIPS 2022.
>
> [6] Esser et al. Scaling Rectified Flow Transformers for High-Resolution Image Synthesis. ICML 2024.
>
> [7] Kondratyuk et al. VideoPoet: A Large Language Model for Zero-Shot Video Generation. ICML 2024.
>
>
> **If you have any more questions, we are happy to discuss them and will do our best to address them!**

---

> > ### Comment · Reviewer_vGqQ · 2025-08-05
> >
> > Thank the author for their response.
> > - I appreciate your contribution on scaling the training of diffusion model training, though I think ViT has its architecture contribution to adapt transformers to vision inputs for the first time and GPT3 is the first work ever showing scaling could push new frontier and in-context learning capabilities with Transformer. Nonetheless, I agree that the contribution of being the first scaling Diffusion LM to this scale is significant and the community will greatly appreciate your comprehensive experiments.
> >
> > - For context length, I understand this is a fundamental limitation for diffusion LMs to not being flexible for the model to decide how long to generate. Would be nice for followup work to remove the need for user to define the length.
> >
> > - Thanks for the explanation on the advantages of removing autoregressive bias.
> >
> > Overall, my questions are well addressed and I will raise my score accordingly.

---

> > > ### Author Response · Authors · 2025-08-06
> > >
> > > We sincerely appreciate your recognition of our work and your decision to raise the score. Following your insightful suggestion, we will further explore, in future work, how to remove the need for users to define the output length hyperparameter. We believe that your constructive feedback has significantly improved both the clarity and the overall quality of the paper.

---

### Official Review · Reviewer_YafR · 2025-07-03

**Clarity:** 3
**Significance:** 4
**Originality:** 4
**Rating:** 6
**Confidence:** 4

**Summary:**

This paper presents LLaDA, a diffusion large language model trained with a masked token prediction objective. The core idea resembles that of MaskGIT or BERT, employing a high and random masking ratio. During inference, the model iteratively predicts masked tokens, selectively retains certain predictions, and re-masks others. Experimental results demonstrate that LLaDA outperforms LLaMA2-7B and achieves performance comparable to LLaMA3-8B.

**Questions:**

1. I find the evaluation procedure of LLaDA somewhat unclear, particularly in how its performance is made comparable to that of autoregressive models. According to Algorithm 3, the evaluation involves Monte Carlo estimation with random masking of tokens. Could the authors clarify whether the target tokens (i.e., the ground-truth answers) are included in the pool of randomly masked tokens, or whether they are always masked? A more detailed explanation of how log-likelihood is computed is preferred.

2. When employing the confidence-based decoding strategy, how much does the generation behavior diverge from the autoregressive paradigm?

3. Given that some proprietary models achieve strong performance on code generation tasks. It would be interesting to know whether LLaDA offers any inference-time advantages over autoregressive models in these datasets, such as HumanEval and MBPP. Has the model been evaluated on such benchmarks, and if so, how does its speed and performance compare?

4. Have you tried to scale a continuous diffusion model instead of the MDM, or say the discrete diffusion models?

**Ethical Concerns:**

["NO or VERY MINOR ethics concerns only"]

**Final Justification:**

The authors' rebuttal solves most of my concerns, except the one about the training efficiency problem compared to ARs. But I think it's not a big problem and I still vote the strong accept of this paper due to the contribution of this paper.

**Quality:**

3

**Strengths And Weaknesses:**

Strengths:

1. This paper successfully scales diffusion language models to the billion-parameter regime, which marks a significant contribution to the community. Demonstrating that an alternative paradigm can achieve performance on par with traditional autoregressive models highlights the potential of non-AR approaches and broadens the design space for future language model development.

2. The performance of LLaDA is impressive. Notably, the model exhibits in-context learning capabilities, which challenge the common belief that such behavior is exclusive to AR models. This suggests that in-context learning may be an emergent property of language, rather than a consequence of autoregressive modeling alone.

3. The authors further fine-tune the model with instruction-tuning and SFT, showing that the pretrained diffusion-based model can match the capabilities of AR models in instruction-following tasks. This underscores the viability of diffusion-based models for practical downstream applications.


Weaknesses:
1. I remain unconvinced that the proposed model truly qualifies as a diffusion model in the conventional sense. While the diffusion perspective offers an interesting theoretical lens for interpreting BERT/MaskGIT-style training, it departs from the standard formulation of diffusion models, particularly those involving continuous-time processes as used in image generation.

2. The training efficiency claims in Figure 3 need further clarification. It is not immediately clear why LLaDA would achieve better training efficiency than autoregressive baselines, especially considering that only a subset of tokens is utilized per iteration. In contrast, autoregressive models typically leverage 100% of the tokens during training. The current results do not convincingly support the claim of higher efficiency.

3. From the inference perspective, the proposed method may be significantly less efficient than autoregressive models, particularly in long-context scenarios. It would be helpful if the authors could provide a direct latency comparison between LLaDA and strong AR baselines such as LLaMA2 or LLaMA3, to better assess the practical deployment feasibility of their approach.

---

> ### Author Rebuttal · Authors · 2025-07-31
>
> # Response to Reviewer YafR
> We thank Reviewer YafR for the recognition of our contributions and the insightful comments. Below is our point-by-point response.
>
>
> ## Q1: Diffusion Formulation
> Thanks for your question. First, we would like to kindly clarify that LLaDA is a **continuous-time but discrete-state** diffusion model (see Eq. (3) in our submission, where the time variable $t$ is continuous). In the first diffusion model paper [1], both continuous-state and discrete-state transition kernels were proposed under the same diffusion formulation (see Table App.1 in Appendix B of [1]). The continuous-state variant is widely known due to its success in image generation, but it currently faces scalability challenges when applied to language tasks (see details in our response to Q7).
>
>
> This has motivated exploration of discrete diffusion language models [2, 3, 4]. Following such a formulation, LLaDA adopts a randomly sampled masking ratio between 0 and 1 (see Lines 82–86), whereas BERT uses a fixed ratio. This fixed masking ratio prevents BERT from being used for sampling and limits its scalability. MaskGIT’s training objective is heuristic and lacks a theoretical link to maximum likelihood, whereas LLaDA’s objective is explicitly derived from maximum likelihood estimation. In particular, MaskGIT's objective misses the $\frac{1}{t}$ term compared to our Eq. (3), further indicating that it is a heuristic formulation.
>
>
> More importantly, **the theoretical insights from diffusion motivated our decision to scale discrete diffusion models for language as a counterpart to autoregressive approaches.** The theoretical analysis of discrete diffusion shows that its training process corresponds to maximum likelihood estimation, which gives us confidence in scaling discrete diffusion to larger models and datasets (as detailed in Lines 17–37). This theoretical foundation also helps explain why LLaDA demonstrates scalability comparable to that of autoregressive models.
>
>
> Given that **LLaDA shares the same theoretical foundation as continuous diffusion models and differs significantly from BERT and MaskGIT in both theoretical origin and practical implementation**, we have adopted the terminology used in the diffusion literature and refer to LLaDA as a diffusion model. We will revise the related work section to make this distinction clearer in the revised version.
>
> [1] Sohl-Dickstein et al. Deep Unsupervised Learning using Nonequilibrium Thermodynamics. ICML 2015.
>
> [2] Austin et al. Structured Denoising Diffusion Models in Discrete State-Spaces. NeurIPS 2021.
>
> [3] Lou et al. Discrete Diffusion Modeling by Estimating the Ratios of the Data Distribution. ICML 2024.
>
> [4] Ou et al. Your Absorbing Discrete Diffusion Secretly Models the Conditional Distributions of Clean Data. ICLR 2025.
>
>
> ## Q2: Insight on Training Efficiency
> Thank you for your question. First, we would like to kindly clarify that Fig. 3 (Page 6) demonstrates that LLaDA exhibits strong scalability, achieving overall performance comparable to the autoregressive baseline across six tasks (as presented in the caption of Fig. 3).
>
> We appreciate your concern that only a portion of the tokens (50% in expectation with a linear schedule) produce loss during training. However, we believe it may be premature to conclude that the data utilization rate of LLaDA is only 50%. Intuitively, although some tokens are not explicitly predicted, they are still processed by the neural network during training.
>
> In practice, even when adjusting the noise schedule to increase or decrease the average masking ratio during training, the final performance remains unaffected. Theoretically, this is because the loss function can be reformulated—via a change of variables—into an equivalent form that is independent of the specific noise schedule (see Sec. 3.4 in [5] and Sec. 3 in [6]). Experimentally, [5] tested different noise schedules, including average masking ratios of 36%, 50%, and $1 - \frac{1 - e^{-10^8}}{10^8} \approx$100%, all of which yielded same loss value (i.e., likelihood) (as detailed in Appendix E.1 of [5]).
>
> Moreover, when modeling the joint distribution of data, autoregressive models optimize only a single factorization order, typically from left to right. In contrast, LLaDA is trained to consider multiple conditioning directions, which may provide greater flexibility and lead to better generalization. The advantage of LLaDA on reversal reasoning is a concrete example of this property.
>
> We will include the above discussion in the experimental section of the revised version.
>
> [5] Sahoo et al. Simple and Effective Masked Diffusion Language Models. NeurIPS 2024.
>
> [6] Shi et al. Simplified and Generalized Masked Diffusion for Discrete Data. NeurIPS 2024.
>
>
>
> ## Q3: Inference Efficiency
> Thank you for your suggestion. We have analyzed the inference efficiency of LLaDA in comparison with LLaMA3 in Fig. 5 (Page 25, with detailed experimental settings described in Lines 863–889).
>
> We employ the widely adopted throughput as the evaluation metric [7]. Compared with latency (i.e., the total time required to generate all tokens) as suggested, throughput measures the number of tokens generated per second and helps avoid discrepancies in output length caused by variations in training data, ensuring a fair comparison.
>
> As shown in Fig. 5 (Page 25), for the GSM8K and Math benchmarks, where the average prompt lengths are 894 and 680 tokens respectively, LLaDA achieves slightly better accuracy than LLaMA3 with KV Cache while delivering 1.5× and 1.8× higher throughput respectively, when configured to sample 4 tokens per step (that is, the number of sampling steps is one-quarter of the generation length). For the HumanEval benchmark, which has a much shorter average prompt length of 132 tokens, LLaDA achieves comparable accuracy and throughput to LLaMA3 with KV Cache even when generating one token per step. However, for the MBPP benchmark, where the average prompt length is 628 tokens, LLaDA lags behind LLaMA3 in accuracy when the throughput is matched.
>
> The above results are obtained without applying KV Cache or advanced parallel sampling techniques to LLaDA. However, recent methods such as these, which are specifically developed for diffusion language models (e.g., [8]), can be applied to LLaDA without retraining to further improve its efficiency.
>
>
> [7] Inception Labs. Mercury: Ultra-Fast Language Models Based on Diffusion. arXiv 2025.06.
>
> [8] Wu et al. Fast-dLLM: Training-free Acceleration of Diffusion LLM by Enabling KV Cache and Parallel Decoding. arXiv 2025.05.
>
>
> ## Q4: Details on Likelihood Evaluation
> Thank you for your question. As detailed in Lines 836–842, following standard practice for autoregressive models, likelihood evaluation is only conducted on multiple-choice tasks. The model computes the conditional likelihood for all candidate options, including the correct answer (without knowing which is correct), and selects the one with the highest likelihood. Accuracy is then used as the evaluation metric. This procedure is both standard and fair.
>
>
> ## Q5: Generation Behavior of the Low-confidence Remasking
>
> Thanks for your question. When using the Low-confidence Remasking strategy, the model tends to generate tokens on the left side of the sequence earlier in a coarse-grained manner, but the generation does not follow a strictly left-to-right autoregressive order.
>
> We selected the first 100 samples from the GSM8K test set and analyzed the position distributions of the 10th, 50th, and 100th generated tokens. The frequency statistics are presented in the tables below. For clarity, we report only the top 10 most frequent positions for each case.
>
> the 10th generated tokens:
> | Position | 12 | 10 | 11 | 13 | 14 | 17 | 16 | 9 | 8 | 6 |
> |----------|----|----|----|----|----|----|----|---|---|---|
> | Freq     |15% |12% |12% |11% | 9% | 7% | 6% |6% |4% |4% |
>
> the 50th generated tokens:
> | Position | 50 | 51 | 52 | 49 | 54 | 48 | 53 | 59 | 56 | 43 |
> |----------|----|----|----|----|----|----|----|---|---|---|
> | Freq     |17% |10% |8% |7% | 6% | 6% | 4% |4% |3% |3% |
>
> the 100th generated tokens:
> | Position | 101 | 100 | 103 | 99 | 112 | 98 | 102 | 105 | 104 | 107 |
> |----------|----|----|----|----|----|----|----|---|---|---|
> | Freq     |13% |11% |10% |9% | 7% | 6% | 6% |5% |5% |4% |
>
>
> ## Q6: Performance and Efficiency on Code Generation
>
> Thank you for your question. We have evaluated LLaDA on both the HumanEval and MBPP benchmarks; please refer to our response to Q3 for details. It is worth noting that our focus is on general performance, and we did not specifically design the training process for code generation. We may consider incorporating a higher proportion of code data or applying common techniques such as reinforcement learning to further improve LLaDA’s code generation capabilities.
>
>
> ## Q7: Scaling Continous Diffusion
>
> This is an excellent question. Our preliminary experiments suggest that, given the same model size and training data, continuous diffusion models fall behind discrete diffusion models. Some existing works [9, 10] in the field have also reported similar observations. We hypothesize that this may be due to a mismatch between discrete data and continuous representations. We believe this is a challenging yet highly impactful direction for future research.
>
> [9] Gulrajani et al. Likelihood-Based Diffusion Language Models. NeurIPS 2023.
>
> [10] Nie et al. Scaling up Masked Diffusion Models on Text. ICLR 2025.
>
> **If you have any more questions, we are happy to discuss them and will do our best to address them!**

---

> > ### Comment · Reviewer_YafR · 2025-08-06
> >
> > Thanks for your detailed response, which solves most of my concerns. I would like to keep my score.

---

> > > ### Author Response · Authors · 2025-08-06
> > >
> > > We greatly appreciate your recognition of our contribution. We believe that your insightful suggestions have helped us improve the quality of the paper. We will incorporate the discussions from the rebuttal phase into the final revision.

---

### Note · Authors · 2025-08-13

We sincerely thank the reviewers, AC, SAC, and PC for their hard work.

Our paper first shows the promise of diffusion models for language modeling at scale and challenges the common assumption that core LLM capabilities, including scalability, in context learning, and instruction following, inherently depend on autoregressive models.

We are pleased that all reviewers indicated our rebuttal has addressed their concerns and that they have no further questions. We greatly appreciate their strong recognition of our contributions, as reflected in comments such as “marks a significant contribution to the community” (Reviewer YafR), “the contribution of being the first scaling Diffusion LM to this scale is significant and the community will greatly appreciate your comprehensive experiments” (Reviewer vGqQ), and “contribution that is somewhat pioneering in nature” (Reviewer APN5).

We will incorporate the discussions from the rebuttal phase into the final version to further improve the quality of our paper. We once again express our sincere gratitude to all reviewers and chairs for their valuable time.

---

### Decision · Program_Chairs · 2025-09-17

**Decision:**

Accept (oral)

**Comment:**

This paper presents a diffusion-based language model scaled to 8 billion parameters and trained on over 2 trillion tokens. The primary contribution is the empirical demonstration that a non-autoregressive architecture can achieve performance comparable to, and in some cases exceeding, well-established autoregressive models of a similar scale (like Llama 3 8B).

All four reviewers are in agreement on the paper's acceptance and identify several key strengths including comprehensive evaluation, strong model performance, and novel capabilities (like addressing the reversal curse)

The initial reviews raised several points for clarification. Reviewers questioned the model's latency and throughput compared to highly optimized autoregressive models with KV caching; they questioned it being called a diffusion model at all. They also asked for more details on handling variable length generation, as the model requires a fixed output length hyperparameter. The reviewers were satisfied by the author response.

For the camera-ready version, the authors are expected to integrate the detailed clarifications and discussions from their rebuttal into the main body of the paper. This includes: (1) the expanded discussion on inference efficiency and throughput comparisons, (2) the clarification of the model's theoretical relationship to, and distinction from, other discrete diffusion models and masked language models, and (3) the discussion on the current approach to variable-length generation and its limitations.

The key reason for my recommendation for oral is the paper's high impact: it establishes a viable, non-autoregressive pathway for foundation models, opening significant new avenues for research